# Strain Hardening, Damage and Fracture Behavior of Al-Added High Mn TWIP Steels

**Manjunatha Madivala [1,*] , Alexander Schwedt [2], Ulrich Prahl [3] and Wolfgang Bleck [1]**

[1] Steel Institute (IEHK), RWTH Aachen University, Intzestraße 1, 52072 Aachen, Germany; bleck@iehk.rwth-aachen.de
[2] Central Facility for Electron Microscopy (GFE), RWTH Aachen University, Ahornstraße 55, 52074 Aachen, Germany; schwedt@gfe.rwth-aachen.de
[3] Institute of Metal Forming (imf), TU Bergakademie Freiberg, Bernhard-von-Cotta-Straße 4, 09599 Freiberg, Germany; Ulrich.Prahl@imf.tu-freiberg.de
* Correspondence: manjunatha.madivala@iehk.rwth-aachen.de; Tel.: +49-241-80-90137

**Abstract:** The strain hardening and damage behavior of Al-added twinning induced plasticity (TWIP) steels were investigated. The study was focused on comparing two different alloying concepts by varying C and Mn contents with stacking fault energy (SFE) values of 24 mJ/m$^2$ and 29 mJ/m$^2$. The evolution of microstructure, deformation mechanisms and micro-cracks development with increasing deformation was analyzed. Al-addition has led to the decrease of C diffusivity and reduction in tendency for Mn-C short-range ordering resulting in the suppression of serrated flow caused due to dynamic strain aging (DSA) in an alloy with 0.3 wt.% C at room temperature and quasi-static testing, while DSA was delayed in an alloy with 0.6 wt.% C. However, an alloy with 0.6 wt.% C showing DSA effect exhibited enhanced strain hardening and ductility compared to an alloy with 0.3 wt.% C without DSA effect. Twinning was identified as the most predominant deformation mode in both the alloys, which occurred along with dislocation glide. Al-addition has increased SFE thereby delaying the nucleation of deformation twins and prolonged saturation of twinning, which resulted in micro-cracks initiation only just prior to necking or failure. The increased stress concentration caused by the interception of deformation twins or slip bands at grain boundaries (GB) has led to the development of micro-cracks mainly at GB and triple junctions. Deformation twins and slip bands played a vital role in assisting inter-granular crack initiation and propagation. Micro-cracks that developed at manganese sulfide and aluminum nitride inclusions showed no tendency for growth even after large deformation indicating the minimal detrimental effect on the tensile properties.

**Keywords:** TWIP steel; deformation twinning; serrated flow; dynamic strain aging; damage; fracture

## 1. Introduction

Through many years of development and application, advanced high strength steels (AHSS), aluminum, magnesium and titanium alloys have proved themselves to be versatile and effective materials for automotive parts [1–6]. However, the demand for safety and cost-effectiveness of the components have raised enormously [7]. Thus it is necessary to develop new materials focusing on the special requirements. Through careful control of chemistry and processing, materials can be tailored to provide optimum performance, keeping an aim on specific applications. By choosing different alloying elements and careful processing, the required microstructural constituents with unique mechanical behavior can be obtained resulting in many types of materials for automotive applications [8]. Thus variations in chemistry, microstructure, and deformation mechanisms are crucial aspects in the design and performance of metallic materials. The current work reveals the

mechanical behavior of the Al-added high Mn twinning induced plasticity (TWIP) steels designed for safety-relevant automotive parts and other applications.

TWIP steels exhibit extraordinary mechanical properties such as exceptional strength (>1000 MPa), high ductility (>50%) and excellent energy absorption capacity (>55 kJ/kg) making them highly suitable for automotive applications [9–15]. The sustained and high strain hardening rate (SHR) of TWIP steels can be ascribed to their deformation mechanisms such as dislocation glide, deformation twinning, and $\varepsilon$-martensite transformation. The materials behavior strongly dependent on their chemical composition, microstructure and deformation conditions [15–19]. The unique characteristic behavior of these steels is the nucleation and propagation of deformation bands during deformation, which results in serrations on the $\sigma$–$\varepsilon$ curves, known as Portevin-Le Chatelier (PLC) effect [20–22]. The serrations in TWIP steels are classified as Type A, depending on their morphology. They are known as locking serrations because of repeated locking and unlocking of dislocations by solute atoms commonly known as dynamic strain aging (DSA) [16,22–24]. It was stated that DSA effectively modified the SHR but its influence on the SHR is limited due to the lower activation energy attained in these steels [21]. Many authors stated that mechanical twinning and its associated dynamic Hall-Petch effect remain the only mechanisms that can be responsible for sustained and high SHR of TWIP steels [10,13,21,25]. However, the influence of DSA on the deformation behavior and its role in manifesting the SHR cannot be ignored. Thus in this study, an alloy showing DSA effect is compared with an alloy without DSA effect. The SHRs of the alloys were correlated to the evolution of twin volume fraction (TVF) with increasing deformation.

The appearance of serrations on the $\sigma$–$\varepsilon$ curves requires the material to be plastically deformed to a certain deformation known as critical strain ($\varepsilon_c$). A detailed study on the X60Mn22 alloy revealed repeated serrations on the $\sigma$–$\varepsilon$ curves at different strain rates and temperatures. With an increase in temperature or grain size, the $\varepsilon_c$ for the initiation of serrations decreased and with the increase of strain rate ($\dot{\varepsilon}$), the $\varepsilon_c$ increased [14,19]. In Fe-Mn-C alloys, it was shown that with the increase in C content from 0.0 wt.% to 1.1 wt.% C, the $\varepsilon_c$ for triggering serrations has decreased [26]. A comparative study on Fe-Mn-C-x alloys with and without Al showed that with Al-addition the $\varepsilon_c$ has increased [27]. It was also shown that with increasing Al content from 0.0 to 3.0 wt.%, the initiation of serrated flow shifted to higher strains or completely suppressed [28]. A similar study also revealed severe serrated flow in an X60Mn22 alloy at different $\dot{\varepsilon}$, whereas in an X60MnAl22-1.5 alloy serrations appeared only at quasi-static $\dot{\varepsilon}$ and the initiation of serrations shifted to high strains [29]. It was shown in X60MnAl18-x alloys with 0.0, 1.5 and 2.5 wt.% Al contents, that severe serrated flow also occurred in Al-containing alloys at elevated temperatures [16]. Thus, it can be clearly understood that $\varepsilon_c$ for the initiation of the serrations in Fe-Mn-C-Al alloys is mainly influenced by stacking fault energy (SFE), the alloying elements such as C and Al, deformation temperature, $\dot{\varepsilon}$ and microstructure of the material. From the experimental observations in the literature, it can be stated that in the alloys of Fe-Mn-C and Fe-Mn-C-Al with different chemical compositions the serrated $\sigma$–$\varepsilon$ behavior can be observed but the range of temperature, $\dot{\varepsilon}$ regimes, microstructure states and the strain $\varepsilon_c$ at which serrations appears could be different. In the current study, an X60MnAl17-1 and X30MnAl23-1 alloys were investigated in detail to study the influence of alloying elements such as C and Al on the serrated flow behavior.

Macroscopic observations revealed an abrupt failure in TWIP steels without any strain localization or necking, which occurred at the intersection of two deformation bands close to the specimen edges. It was claimed that the increased stress concentration caused by the intercepting deformation twins and the slip band extrusions at grain boundary (GB) has led to micro-crack formation. Inter-granular micro-crack initiation and propagation events were observed in the microstructure at GBs and triple junctions along with the rapid nucleation of minute voids [14]. A similar study also showed micro-cracks at the intersection of deformation bands mainly at the edge and side surfaces of the tensile specimen. The cracks were observed at GB junctions and the mechanical twin boundaries [30]. It was claimed that the exponential increase of the macroscopic void volume fraction led to the sudden failure or decrease of SHR in TWIP steels [31]. A detailed study on the evolution of damage in TWIP steel by 3D X-ray tomography experiments has shown that the average void diameter and the stress

triaxiality remained constant throughout the deformation until failure. It was claimed that the damage development process involved rapid nucleation of minute voids combined with substantial growth of the largest voids [32]. The distribution of tiny voids with few elongated cavities close to the fracture surface in shear tests clearly indicated localized failure [33]. The above-mentioned studies were on TWIP steels showing DSA effect, where an abrupt failure occurred in the material. However, the addition of Al not only increases ductility but also changes the local deformation and fracture behavior. Thus in this study damage and failure behavior of Al-added TWIP steels were investigated both at the micro- and macro-scale. The current work focuses on the relationship between the deformation mechanisms and the mechanism of micro-crack formation.

In the current study, the strain hardening, damage and fracture behavior of Al-added TWIP steels with different chemical compositions were investigated. To ascertain the role of the interstitial C atoms on the interaction between Mn-C short-range ordering (SRO) and dislocations in causing DSA phenomena, two alloys with different C contents were studied. By analyzing the local plastic strain ($\varepsilon_{local}$) evolution, the serrated flow behavior caused due to the nucleation and propagation of deformation bands and their subsequent influence on the failure initiation during the deformation was investigated. The measurement of a rise in temperature during deformation aided in the accurate estimation of SFE, which enabled the prediction of change in deformation mechanisms. The evaluated TVF and the predicted SFE was used to explain the SHR, the mechanical behavior of alloys. The evolution of microstructure, the mechanism of micro-crack initiation and their development with increasing strain were studied in detail. The influence of non-metallic inclusions on the mechanical properties and micro-crack formation was also investigated.

## 2. Materials and Experimental Methods

The materials investigated in this study are Al-added high Mn TWIP steels produced in the laboratory, designated as X60MnAl17-1 and X30MnAl23-1. The electron backscatter diffraction (EBSD) technique was used to characterize the initial microstructure such as the grain size and the evolution of twinning with increasing strain. The optical and field-emission scanning electron microscope (SEM) with energy dispersive X-ray spectroscopy (EDS/EDX) was used to qualitatively and quantitatively analyze the inclusions type, size, and distribution. Interrupted micro-tensile tests were carried out in SEM to study the evolution of microstructure and micro-cracks development during deformation. Tensile tests were carried out in conjunction with the digital image correlation (DIC) to investigate the macroscopic material behavior. An infrared thermography camera along with video extensometer was used to measure the temperature rise due to adiabatic heating (AH) during deformation. The elastic properties of the material were determined by the ultrasonic method. The fracture behavior of the macroscopic tensile specimens was analyzed in the SEM.

### 2.1. Materials Processing

The materials were produced by ingot-casting using a vacuum induction furnace. The cast ingots (each ∼30 kg) were homogenized in a muffle furnace at 1150 °C for 5 h, in order to reduce the segregation of alloying elements, especially Mn. The homogenized ingots (140 mm in height) were then forged at 1150 °C to a height of 55 mm, followed by another homogenization and then hot rolled at 1150 °C to a sheet thickness of 2.8 mm. The hot rolled sheets were cold-rolled to 50% reduction to get a final sheet thickness of 1.4 mm. Finally, annealing heat treatment was carried out in a salt bath furnace at 900 °C for 20 min, followed by quenching in water to obtain a completely recrystallized microstructure. The major differences in alloy compositions are C and Mn contents. The alloy X60MnAl17-1 has 0.6 wt.% C, 17.17 wt.% Mn, whereas alloy X30MnAl23-1 has 0.3 wt.% C, 22.43 wt.% Mn. The chemical compositions and SFE values of the alloys are presented in Table 1. The chemical composition was determined using optical emission spectroscopy. The Mn content from 0.001 to 20 wt.% with an accuracy of ±0.05, C from 0.001 to 1.2 wt.% with an accuracy of ±0.01 and Al from 0.001 to 2 wt.% with an accuracy ±0.01 can be determined. The Mn content above

20 wt.% is verified by melt extraction analysis. The SFE was calculated using the sub-regular solution thermodynamic model of Saeed-Akbari et al. [11]. The interface energy ($\sigma^{\gamma/\varepsilon}$) value of 10 mJ/m$^2$ as recommended in [11,34] was used for calculating the SFE. The model is implemented as a MATLAB$^®$ program to evaluate the SFE as explained in [19]. Based on their chemical composition, the evaluated SFE value of X60MnAl17-1 and X30MnAl23-1 are 29 and 24 mJ/m$^2$ respectively.

**Table 1.** Chemical compositions (in wt.%) and the stacking fault energy (SFE) values (in mJ/m$^2$) of the investigated alloys.

| Alloy | C | Si | Mn | P | S | Cr | Ni | Cu | Al | V | N | Fe | SFE |
|-------|------|------|-------|---------|---------|------|------|------|------|------|-------|------|-----|
| X60MnAl17-1 | 0.60 | 0.06 | 17.17 | <0.009 | <0.006 | 0.05 | 0.04 | 0.03 | 1.50 | 0.07 | 0.015 | Bal. | 29 |
| X30MnAl23-1 | 0.30 | 0.04 | 22.43 | <0.009 | <0.005 | 0.05 | 0.04 | 0.02 | 1.39 | 0.10 | 0.013 | Bal. | 24 |

## *2.2. Microstructure Characterization*

### 2.2.1. Micro-Tensile Tests

The un-deformed and deformed microstructures were analyzed by using EBSD on the RD-TD plane (RD/TD: rolling/transverse direction). The micro-tensile test specimens with a gauge length of 2 mm, a width of 1 mm and having a thickness of 1.4 mm as shown in Figure 1 were obtained by wire erosion cutting from the cold-rolled and annealed sheet. The samples were initially mechanically ground and then polished up to 1 μm using diamond suspension. They were further electro-polished by applying a voltage potential of 35 V for 15 s using an electrolyte consisting of 95% acetic acid and 5% perchloric acid. A field-emission JEOL JSM-7000F SEM equipped with an EDAX-TSL Hikari detector operating at a voltage of 20 kV and a probe current of about 20–30 nA was used for the measurements. The data was analyzed using TSL OIM$^®$ Analysis 7 software. A step size of 100 nm was chosen for the measurements and all the scanned points with a confidence index (CI) value of below 0.1 were not considered for the analysis. The measurements were carried out in the middle of the sample as shown in Figure 1c and at a true strain of 0.0, 0.1, 0.2, 0.3, 0.4 and $\varepsilon_{\text{necking}}$. The procedure used for processing EBSD data to identify the grains and twins is described in [14,19]. The TVF was evaluated based on the relative fraction of the pixel areas of Σ-3 twins to the measurement areas at different macroscopic strains [19].

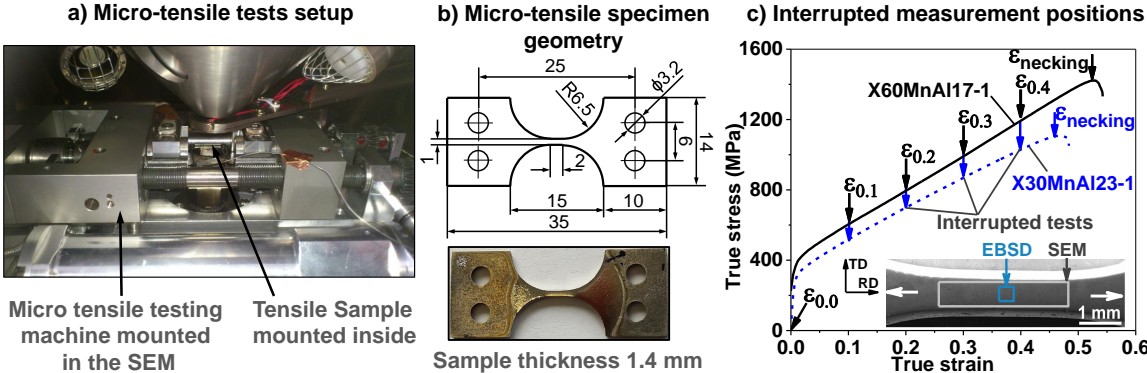

**Figure 1.** In-situ micro-tensile tests in conjunction with scanning electron microscope (SEM) and electron backscatter diffraction (EBSD): (**a**) test setup, (**b**) specimen geometry, (**c**) true stress-strain curves obtained for X60MnAl17-1 and X30MnAl23-1; tests were carried out with a crosshead displacement rate of 0.2 mm/min, which corresponds to a quasi-static $\dot{\varepsilon}$ of about 0.001 s$^{-1}$ and interrupted at different strains as pointed to observe the microstructure and micro-crack development; EBSD measurement area and the region of SEM investigations on the micro-tensile specimen is shown; white arrows indicate tensile loading direction.

The micro-tensile tests were carried out in a field emission SEM to investigate the mechanisms of micro-crack formation. The tests were interrupted at different strains to observe the micro-crack initiation and their development as shown in Figure 1. The tests were performed at room temperature (RT) and by applying a crosshead displacement rate of 0.2 mm/min which corresponds to a quasi-static $\dot{\varepsilon}$ of 0.001 s$^{-1}$ using an in-situ micro-tensile testing device mounted in the SEM. The macroscopic fractured samples were also analyzed in SEM to study the failure behavior of the alloys. To observe the changes in dimple size and morphology at the inclusions several SEM images were taken at different locations. The average dimple size was determined by image processing as explained in [19].

### 2.2.2. Inclusions and Fracture Surface Analysis

The specimens of size $20 \times 20$ mm$^2$ (RD-TD plane) were embedded in an epoxy resin, ground and mechanically polished with a diamond suspension up to 1 μm for inclusions analysis. They were characterized quantitatively to determine the size and area fraction according to DIN EN 10247 standard by optical microscopy in conjunction with an image-analysis software. Due to the inherent high-contrast ratio between non-metallic inclusions and the matrix, inclusions were accurately determined by image-analysis and quantitative data was extracted. The inclusions were analyzed qualitatively based on their chemical composition by SEM/EDS technique.

### 2.3. Mechanical Tests

### 2.3.1. Macro-Tensile Tests with Digital Image Correlation (DIC)

Macroscopic tensile tests were carried out according to DIN EN ISO 6892 standard. A sample geometry of gauge length 75 mm, a width of 12.5 mm and having a thickness of 1.4 mm as shown in Figure A1 was used. The samples were obtained by water-jet cutting along the RD and the machined edges were polished up to a surface roughness of 0.125 μm. The tests were performed at RT in conjunction with DIC or video extensometer using a Zwick/Roell Z100 machine with a crosshead displacement rate of 1.8 mm/min, which corresponds to a quasi-static $\dot{\varepsilon}$ of about 0.001 s$^{-1}$. A force transducer was used to measure force and a video extensometer was used to measure the elongation. The GOM Aramis 12M 2D DIC system (see Figure A1) was used for evaluating the $\varepsilon_{local}$ distribution and to investigate the initiation and propagation of deformation bands during straining. A high contrast black and white stochastic pattern was prepared on the samples for the measurements. The images were captured at a rate of 1 Hz during the test and a facet size of 100 μm was set for the $\varepsilon_{local}$ evaluation. The force data was imported into the ARAMIS DIC software for calculating the $\sigma$–$\varepsilon$ curves. The mechanical properties were evaluated following the ASTM E517, ASTM E646, and DIN ISO 10275 standards. The properties evaluated using video extensometer and DIC were almost similar. The SHR curves were evaluated from the smoothened $\sigma$–$\varepsilon$ curves.

### 2.3.2. Macro-Tensile Tests with High-Speed Thermocamera

The rise in temperature due to adiabatic heating during deformation was measured by high-speed infrared thermography camera of type varioCAM® hr along with video extensometer as shown in Figure A2. The temperature measurement of the camera is from $-40$ °C to 1200 °C. The overall accuracy of temperature measurement within the range of calibration is $\pm 1$ °C. A tensile sample with a gauge length of 30 mm and a width of 6 mm as shown in Figure A2 was tested using a Zwick/Roell Z100 machine at a $\dot{\varepsilon}$ of 0.001 s$^{-1}$. The emissivity was also calibrated before the measurements and the specimen was coated with black lacquer to minimize the reflections from the surroundings. The images were acquired at a frequency of 1 Hz in full-frame-mode with a maximum of $384 \times 288$ pixels resolution using the IRBIS® online software. The data was analyzed to extract the temperature variation within the gauge length and co-related with the strain measured from the video extensometer.

### 2.3.3. Elastic Constants Measurement

Elastic properties of the alloys were determined by using the GE USM35 ultrasonic testing machine. Longitudinal and transversal pulses were measured by using normal incident 20 MHz (CLF4) longitudinal and 10 MHz (K7KY) shear transducers respectively. The presence of air can disrupt the propagation of the acoustic waves into the material, therefore a viscous liquid such as oil (longitudinal)/honey (transversal) was applied as contact material between the transducer and the sample surface. With the accurate measurement of the sample thickness ($l$) and the transit time ($t$) between the peaks of consecutive echoes, the longitudinal ($v_l$) and transversal ($v_t$) sound velocities of the material was calculated by using $v_{l/t} = l/(\frac{t}{2})$. The density ($\rho$) of the alloys was determined by using AccuPyc 1330 pycnometer. The elastic properties of the alloys such as Young's modulus ($E$), shear modulus ($G$) and Poisson's ratio ($\nu$) were calculated using the relations given by [35]

$$E = 3\rho v_t^2 (v_l^2 - \frac{4}{3}v_t^2)/(v_l^2 - v_t^2), \qquad G = \rho v_t^2, \qquad \nu = \frac{1}{2}(v_l^2 - 2v_t^2)/(v_l^2 - v_t^2) \tag{1}$$

## 3. Results

### 3.1. Microstructure Analysis

The initial microstructure of both alloys X60MnAl17-1 and X30MnAl23-1 consisted of fully austenitic ($\gamma$) grains with an average grain size of 16.0 µm and 12.0 µm respectively. The inverse pole figure (IPF) maps with respect to the RD depicting the grain sizes and the crystal orientation distribution are shown in Figure 2a,b. It can be observed from Figure 2d that for an alloy X30MnAl23-1, the grain size varies within a range of about 1–40 µm, with small fractions of grains larger than 25 µm, whereas for an alloy X60MnAl17-1 shown in Figure 2c, the grain size varies with a large range of about 1–50 µm, with small fractions of grains larger than 35 µm. The initial texture is rather weak for both alloys with a texture index value of 1.33 for X60MnAl17-1 and 1.18 for X30MnAl23-1, as shown in Figure 2a,b.

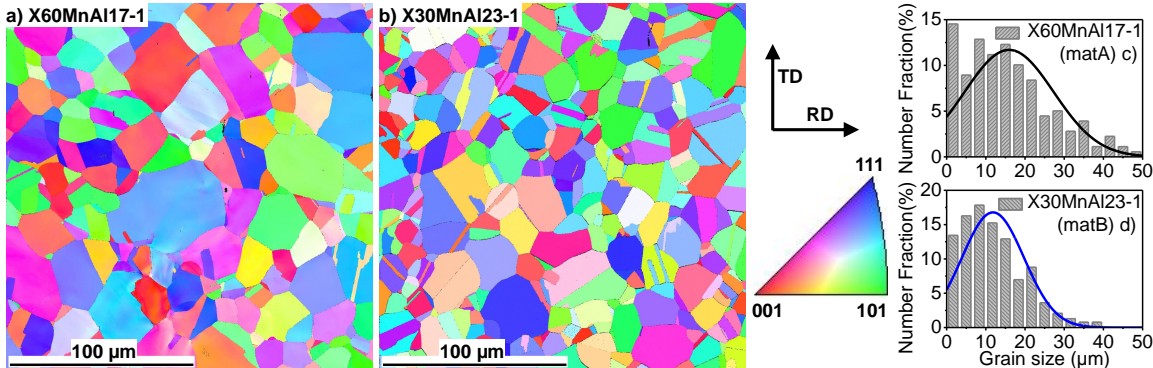

**Figure 2.** The EBSD inverse pole figure (IPF) maps of the undeformed samples showing grain size and orientations: (**a**) X60MnAl17-1, (**b**) X30MnAl23-1. Grain size fitted using normal distribution function shown in: (**c**) X60MnAl17-1, (**d**) X30MnAl23-1; microstructures of both alloys is completely austenite ($\gamma$); the average grain size of X60MnAl17-1 is 16.0 µm and X30MnAl23-1 is 12.0 µm.

The estimated area fraction and the size of inclusions present in the materials are presented in Table 2. The measured area fraction of inclusions in both alloys is ∼0.1%. The average size of the inclusions in X60MnAl17-1, X30MnAl23-1 is about 1.2 µm and 1.5 µm respectively. The distribution of different type of inclusions present in the materials is shown in Figure 3. EDS analysis performed on the inclusions at different locations indicate the presence of manganese sulfides (MnS) and aluminum nitrides (AlN), both in globular and elongated shapes as shown in Figure 2. The globular MnS inclusions have slightly high fractions of Mn and S contents compared to the base material, whereas the elongated MnS contains a higher fraction of Mn and S.

**Table 2.** Cleanliness analysis: area fraction, average and largest size of the inclusions. Size of the smallest inclusions found in both alloys is ∼0.5 µm. The scatter was estimated by analyzing four to five images taken at different locations.

| Alloy | Area Fraction (%) | Average Size (µm) | Largest Size (µm) |
|---|---|---|---|
| X60MnAl17-1 | 0.10 ± 0.01 | 1.23 ± 0.05 | 8.0 ± 1.5 |
| X30MnAl23-1 | 0.11 ± 0.02 | 1.50 ± 0.10 | 12.0 ± 2.0 |

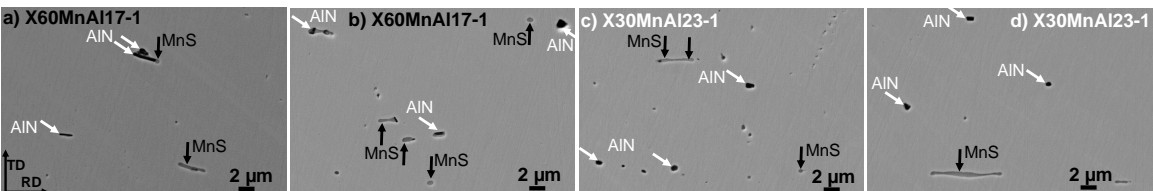

**Figure 3.** SEM micrographs depicting non-metallic inclusions present in the investigated alloys: X60MnAl17-1 (**a,b**); X30MnAl23-1 (**c,d**); manganese sulphide (MnS) (black arrows) and aluminium nitride (AlN) (white arrows).

*3.2. Mechanical Properties*

The mechanical properties of the TWIP steels vary with chemical composition due to change in SFE resulting in different deformation behavior. The stress-strain curves and the SHR for X60MnAl17-1 and X30MnAl23-1 obtained by uni-axial tensile tests at RT are shown in Figure 4. The $\sigma$–$\varepsilon$ curve of X30MnAl23-1 exhibit homogenous flow behavior throughout the deformation, whereas periodic serrations of type A can be observed in the flow curve of X60MnAl17-1. Both X60MnAl17-1 and X30MnAl23-1 show linear flow behavior and slight necking just before the final failure. Initially, strain hardening for both alloys is characterized by a sharp drop of the SHR and then it increases to reach a constant value with small increase in deformation. Thereafter, both alloys show a general drop with a pronounced multistage character at different strains. The first marked inflection in SHR after the sudden drop occurs at ∼0.04 true strain for X60MnAl17-1 and at ∼0.03 true strain for X30MnAl23-1 to attain the highest point of strain hardening. The peak value of SHR for X60MnAl17-1 is about 2460 MPa and X30MnAl23-1 is about 2360 MPa and after that SHR started to decrease steadily for both alloys. The second marked inflection in SHR occurs at ∼0.15 true strain for X60MnAl17-1 and at ∼0.18 true strain for X30MnAl23-1. The SHR increases at second inflection for X60MnAl17-1 compared to a steady decrease in SHR for X30MnAl23-1 until failure. The SHR of X60MnAl17-1 attains another maximum between 0.3 and ∼0.4 true strain, and then it decreases continuously until failure. The SHR for both alloys at the beginning of the deformation is above 2350 MPa and it decreases to about 1400 MPa at failure. Since the SHR of both alloys shows multistage hardening behavior with increasing macroscopic strain indicating a change in active deformation mechanisms at different strains. The serrated flow caused due to DSA in X60MnAl17-1 alloy after a strain of ∼0.15 could have played a crucial role in enhancing the strain hardening and ductility of the material.

The variation in the mechanical properties of X60MnAl17-1 and X30MnAl23-1 are presented in Table 3. The X60MnAl17-1 with 0.6 wt.% C has a higher yield strength of about 294 MPa compared to 246 MPa of X30MnAl23-1 with 0.3 wt.% C. Similarly, the tensile strength of X60MnAl17-1 and X30MnAl23-1 are very different, which are about 844 MPa and 693 MPa respectively. The TE of X60MnAl17-1 is 70%, which is larger compared to 63% of X30MnAl23-1. The density of X60MnAl17-1 is slightly lower and E, G are much larger compared to X30MnAl23-1. The mechanical properties of X60MnAl17-1 and X30MnAl23-1 presented in Table 3 are very different even though both the designed alloys are very similar in terms of SFE values, heat treatment, and microstructure.

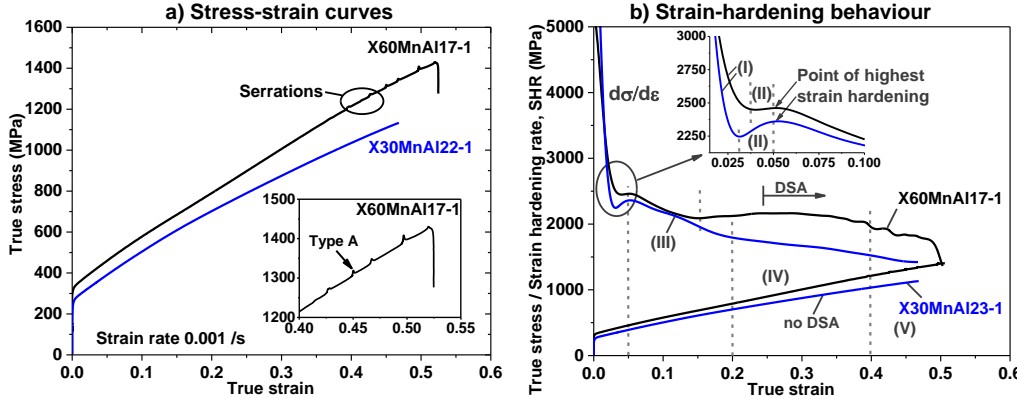

**Figure 4.** Macroscopic material response: (**a**) True stress-strain curves (**b**) Strain-hardening behavior.

**Table 3.** Mechanical properties of Al-added TWIP steels: Yield/ultimate tensile strength (YS/UTS), uniform/total elongation (UE/TE), Lankford coefficient (r-value), strain hardening exponent (n-value), density ($\rho$), Young's modulus (E), shear modulus (G) and Poisson's ratio ($\nu$). All data are average values determined from at least three parallel experiments.

| Alloy | YS | UTS | UE | TE | r-Value | n-Value | $\rho$ | E | G | $\nu$ |
| Unit | MPa | MPa | % | % | - | - | kg/m$^3$ | GPa | GPa | - |
|---|---|---|---|---|---|---|---|---|---|---|
| **X60MnAl17-1** | $294 \pm 10$ | $844 \pm 15$ | $65 \pm 5$ | $70 \pm 5$ | $0.90 \pm 0.01$ | $0.35 \pm 0.01$ | $7700 \pm 10$ | $188 \pm 2$ | $75 \pm 1$ | $0.267 \pm 0.01$ |
| **X30MnAl23-1** | $246 \pm 10$ | $693 \pm 15$ | $62 \pm 2$ | $63 \pm 2$ | $0.83 \pm 0.01$ | $0.37 \pm 0.01$ | $7715 \pm 5$ | $161 \pm 1$ | $63 \pm 1$ | $0.274 \pm 0.01$ |

### 3.3. Deformation Mechanisms

The chemical composition of an alloy has a significant influence on the SFE, which controls the activation of different deformation mechanisms during deformation. Hence SFE of an alloy plays a crucial role in the evolution of microstructure and its phases. The SFE value of 29 mJ/m$^2$ for X60MnAl17-1 and 24 mJ/m$^2$ for X30MnAl23-1, indicate that both alloys are designed to exhibit twinning induced plasticity in combination with dislocation glide as a major deformation mechanism. Activation of DSA in an X60MnAl17-1 alloy can be observed from Figure 4, whereas in an X30MnAl23-1 alloy DSA is not activated.

The evolution of microstructure and deformation mechanisms is shown in Figure 5. Annealing twins can be observed in the initial stages of deformation as shown in Figure 5a,e (black arrows). Mechanical twins or twin bundles could not be observed in EBSD image quality maps in both alloys at a strain of ~0.1. However, an advanced technique such as transmission electron microscopy is essential for observing the initiation of twins. With the small increase in macroscopic strain to ~0.2, mechanical twins can be observed in the grains oriented along <111> (see Figures 5b,f and 13). It could be observed that the fraction of twinned grains or twin area fraction in an X60MnAl17-1 alloy is much high compared to an X30MnAl23-1 alloy. Increasing the deformation further to ~0.3 lead to further nucleation and growth of mechanical twins (white arrows). More intense twinning can be observed in grains of an X60MnAl17-1 alloy compared to an X30MnAl23-1 alloy. The grains which are oriented along the <101> direction reorient toward the <111> direction during deformation and twins initiate in these grains as shown in Figure 13. At a macroscopic strain of ~0.4, both primary and secondary twins can be observed in both alloys. In Figure 5d,h, mechanical twins can be observed in most of the grains indicating large deformation at micro-scale. The twin bundles can be very well identified and the increase in density of mechanical twins can be seen with increasing macroscopic strain. At a strain of ~0.4, the mechanical twins are saturated in the grains in both alloys with no scope for further twinning. The area fraction of detected twins or twin bundles with increasing macroscopic strain is shown in Figure 13. The intense twinning in the grains can be observed before the failure initiation. High TVF in an X60MnAl17-1 alloy has led to better SHR and increased ductility compared to an X30MnAl23-1 alloy. There are many instances in the microstructure where twinning occurs in both

the neighboring grains and share a common GB (light blue arrows). The heavily deformed grains can also have a common junction (yellow arrow). The stress level in such GBs and triple junctions is very high compared to other grains. Such high stressed regions of the microstructure are prone to initiation of micro-cracks.

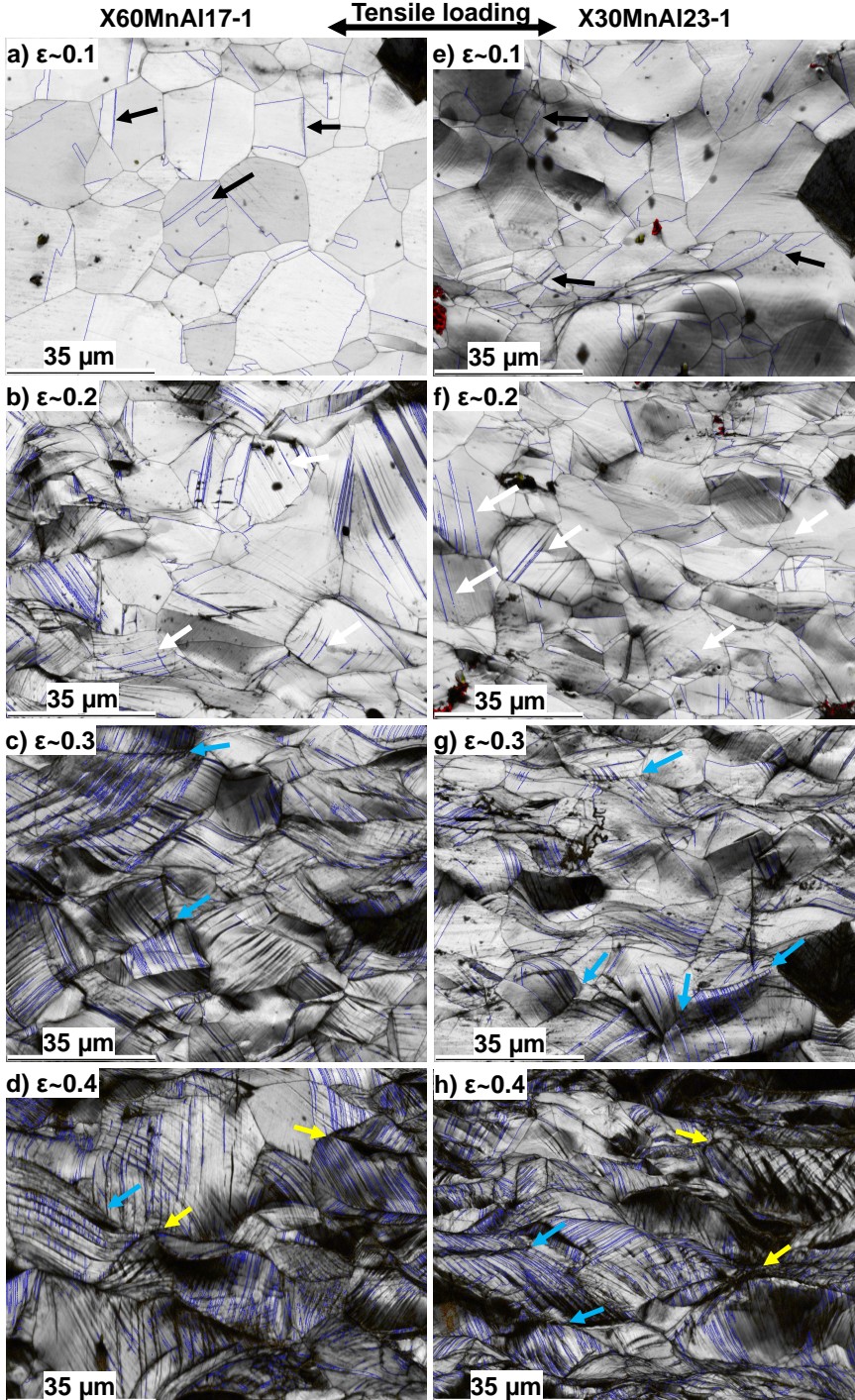

**Figure 5.** EBSD image quality (IQ) maps depicting the evolution of twinning with increasing macroscopic strain: X60MnAl17-1 (**a**) $\varepsilon_{\sim 0.1}$ (**b**) $\varepsilon_{\sim 0.2}$ (**c**) $\varepsilon_{\sim 0.3}$ (**d**) $\varepsilon_{\sim 0.4}$ (left column), X30MnAl23-1 (**e**) $\varepsilon_{\sim 0.1}$ (**f**) $\varepsilon_{\sim 0.2}$ (**g**) $\varepsilon_{\sim 0.3}$ (**h**) $\varepsilon_{\sim 0.4}$ (right column); blue color in IQ maps in (**a**,**e**) indicates annealing twin boundaries and detected $\Sigma$-3 deformation twin boundaries in (**b**–**d**,**f**–**h**); The arrows indicate annealing twin boundaries (black), nucleation of deformation twins (white), intercepting twins at grain boundary (light blue) and triple junctions (yellow); pixels in black are unindexed points.

### 3.4. Local Deformation Behavior

The chemical composition of the alloy mainly C content, has a significant influence on the local deformation behavior. The $\sigma$–$\varepsilon$ behavior extracted from the quasi-static tensile tests at RT carried out in conjunction with DIC is shown in Figure 6a. The serrations can be observed on the $\sigma$–$\varepsilon$ curve caused due to DSA in an X60MnAl17-1 alloy, whereas smooth $\sigma$–$\varepsilon$ curve without any serrations in an X30MnAl23-1 alloy. The DSA in X60MnAl17-1 alloy has led to the plastic instability in the form of initiation and propagation of the deformation bands. The deformation bands initiated at a macroscopic strain of $\sim$24% during deformation propagate until failure. The strain at which deformation bands initiated is shown in Figure 7I and the region at which deformation bands were active is indicated by a rectangle in Figure 6a. After a strain of $\sim$24%, deformation bands nucleated steadily and continuously. The plateau between the two consecutive serrations peaks corresponds to the nucleation and propagation of deformation band within the gage length. The serration peak corresponds to the initiation/disappearance of the deformation band outside the gage length. The initiation and propagation of deformation bands during deformation for X60MnAl17-1 is shown in Figure 7i–k. The enlarged view of $\sigma$–$\varepsilon$ curve from Figure 6a, where deformation bands were active is shown in Figure 7I. It can be observed that stress jumps at serrations at the beginning are smaller compared to the final stages of deformation. The deformation bands nucleated at one shoulder end or in the middle of the specimen propagate to the other end. For up to $\sim$28% strain, only one deformation band nucleated and propagated, whereas after 28% strain, two intersecting deformation bands nucleated and propagated during the deformation (see Figure 7j,k). It is clear from the local strain rate ($\dot{\varepsilon}_{local}$) distribution in Figure 7i–k that, the deformation is localized in the deformation bands, whereas it is uniform in other regions. During initiation of deformation bands, the $\varepsilon_{local}$ accumulation in deformation bands is quite low, but in the subsequent bands, large strain localization can be observed (see Figure 7l–n). The $\varepsilon_{local}$ is larger in the regions where deformation band has already passed compared to the region where deformation band is moving towards or yet to pass through. However, the difference in $\varepsilon_{local}$ between the regions where the deformation band already passed by compared to the regions where the deformation band moving towards is $\leq$5% strain. Thus nucleation and propagation of deformation bands during deformation has resulted in inhomogeneous deformation behavior in X60MnAl17-1 alloy. The velocity at which deformation bands propagate decrease exponentially with increasing strain as shown in Figure 7II. The band velocity at the beginning is $\sim$2 mm/s, which is drastically reduced to $\sim$0.5 mm/s in the end. The failure in X60MnAl17-1 alloy occurred at the intersection of deformation bands when their motion is hindered significantly as shown in Figure 7k,n. The localization of strain within the deformation bands is larger compared to the $\varepsilon_{local}$ distribution in other regions. Hence strain localization caused at the deformation bands influenced the failure initiation in X60MnAl17-1 alloy. For X30MnAl23-1 alloy, the $\dot{\varepsilon}_{local}$ distribution is shown in Figure 7u–w and $\varepsilon_{local}$ distribution is shown in Figure 7x–z. It could be observed that the $\varepsilon_{local}$ and $\dot{\varepsilon}_{local}$ is quite uniform until the beginning of necking. The necking begins at a macroscopic strain of $\sim$60% and with further straining failure occurs due to strain localization. Even though both alloys deform by twinning in combination with dislocation glide, the local deformation behavior in X30MnAl23-1 alloy is homogenous compared to inhomogeneous behavior in X60MnAl17-1 alloy due to the activation of DSA.

The rise in temperature due to adiabatic heating could influence the local deformation behavior. The temperature distribution over the entire gage length of the specimen at different elongations is shown in Figure 8b,d and the temperature variation in the middle of the specimen is shown in Figure 8a,c. From Figure 8, it can be observed that the temperature of the specimen rises due to adiabatic heating (AH) with increasing strain. The temperature distribution is inhomogeneous in X60MnAl17-1 alloy due to the propagation of deformation bands compared to the homogenous temperature distribution in X30MnAl23-1 alloy. The temperature increased above RT by 14 °C in an X60MnAl17-1 alloy and by 12 °C in an X30MnAl23-1 alloy. Thus AH has resulted in an increase of SFE from 24 to 25 mJ/m$^2$ in an X30MnAl23-1 alloy and from 29 to 31.5 mJ/m$^2$ in an X60MnAl17-1 alloy. However, the SFE increase due to AH is not significant for the change in deformation mechanisms.

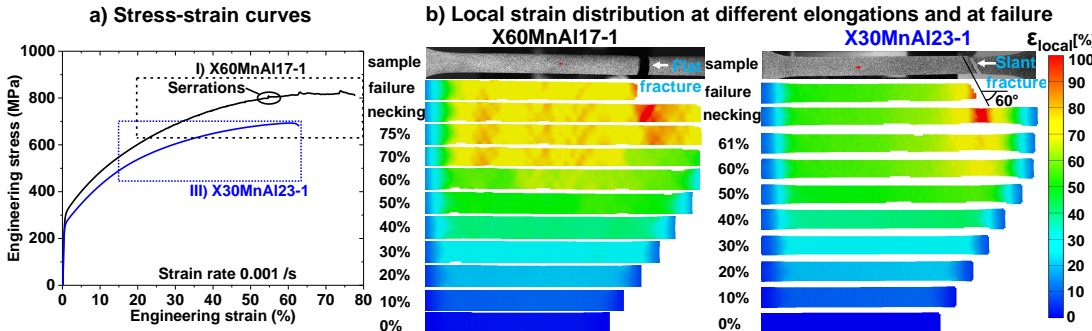

**Figure 6.** (**a**) Stress-strain behavior extracted from the macroscopic tensile tests carried out in conjunction with digital image correlation (DIC). (**b**) Local strain distribution: X60MnAl17-1—Failure at the intersection of deformation bands with large strain localization (see also Figure 7k), X30MnAl23-1—Failure due to diffuse necking and large strain localization; rectangles indicate regions chosen to investigate deformation bands; left: macroscopic strain; right: legend for local von Mises effective plastic strain ($\varepsilon_{local}$) distribution.

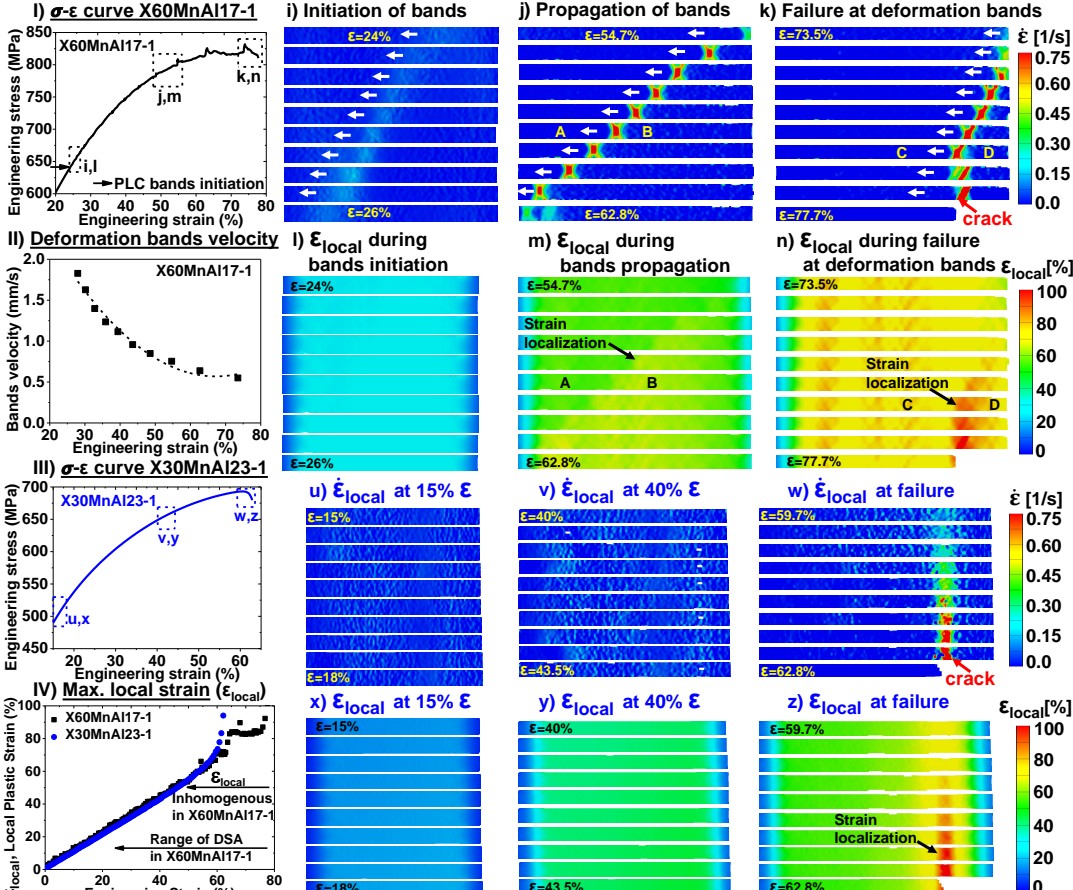

**Figure 7.** Deformation bands initiation, propagation and their location at the point of failure obtained from the tensile tests in conjunction with DIC: (**I**) X60MnAl17-1: enlarged view of $\sigma$–$\varepsilon$ curve depicting the initiation of deformation bands (**i,l**), propagation of deformation bands (**j,m**) and failure at deformation bands (**k,n**), (**II**) plot of the deformation bands velocity versus strain, (**III**) X30MnAl23-1: enlarged view of $\sigma$–$\varepsilon$ curve depicting no deformation bands initiation and propagation (**u,v,x,y**) and failure due to diffuse necking and strain localization (**w,z**), (**IV**) plot of the maximum local plastic strain versus strain; (**i–k,u–w**) shows local strain rate ($\dot{\varepsilon}_{local}$) distribution; (**l–n,x–z**) shows local strain ($\varepsilon_{local}$) distribution; the legend for $\dot{\varepsilon}_{local}$ and $\varepsilon_{local}$ distribution (**right**); white arrows indicate the direction of propagation of the deformation bands; the macroscopic strain at the start and end of deformation band is given; images were shown in steps of ∼0.4–0.6% macroscopic strain.

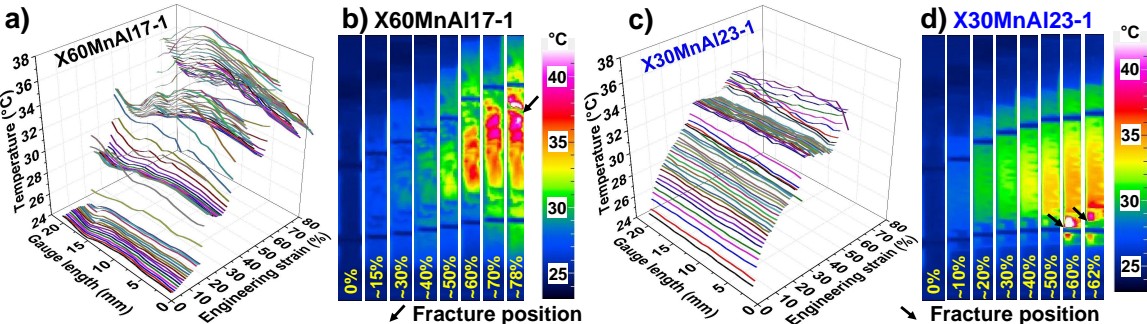

**Figure 8.** Temperature increase due to adiabatic heating in the macroscopic tensile tests carried out in conjunction with infrared thermography camera and video extensometer: (**a**,**c**) Rise in surface temperature during deformation, (**b**,**d**) Distribution of local temperature at different elongations and at failure.

The $\varepsilon_{local}$ distribution along the tensile direction at different elongations is shown in Figure 6b. It can be observed that over the entire gage length of the tensile specimen, the $\varepsilon_{local}$ distribution is uniform until 50% elongation. In an X60MnAl17-1 alloy, inhomogeneous distribution of $\varepsilon_{local}$ can be observed due to deformation bands propagation. The failure initiated at the point of intersection of deformation bands close to the edges of the tensile specimen. In an X30MnAl23-1 alloy for the elongations above 50%, the strain starts to localize within the gage length of the tensile specimen. The region of strain localization has increased further with the increase in deformation and then finally leading to abrupt failure in the localized region.

Slant fracture with pronounced necking can be observed on the broken tensile sample. The macroscopic failure strain ($\varepsilon_f$) in an X60MnAl17-1 alloy is ~77%, whereas the local failure strain ($\varepsilon_{f\_local}$) in the deformation band is ~98%. Similarly in an X30MnAl23-1 alloy $\varepsilon_f$ is ~64% and the $\varepsilon_{f\_local}$ is ~98%. This large difference between the $\varepsilon_{f\_local}$ and the $\varepsilon_f$ indicates that a large amount of strain localization occurred before the final failure. The major differences in failure initiation can be observed in the alloys with and without DSA effect. In an X60MnAl17-1 alloy with DSA effect, failure occurred at the intersection of deformation bands close to edges, whereas in an X30MnAl23-1 alloy without any DSA effect, failure occurred by classical necking and strain localization.

### 3.5. Mechanisms of Damage and Failure

The damage initiation is mainly influenced by the active deformation mechanisms and the presence of non-metallic inclusions in the material. The evolution of microstructure with increasing strain is shown in Figure 9. The micro-crack formation events in the microstructure are shown in Figure 10 and at inclusions is shown in Figure 11. The damage development in both the alloys could be explained in three stages namely damage incubation stage ($\varepsilon_{0.10-0.3}$), damage nucleation stage ($\varepsilon_{0.3-necking}$) and damage growth stage ($\varepsilon_{necking-failure}$). In an X60MnAl17-1 alloy, the major active deformation mechanisms were twinning and slip, activation of DSA and occurrence of intense twinning can be observed at lower strains, hence the damage nucleation begins at earlier stages of deformation. Whereas in an X30MnAl23-1 alloy, only twinning and slip were the active deformation mechanisms and the occurrence of intense twinning can be observed at later stages of deformation close to necking, which could have delayed the beginning of damage nucleation.

During the damage incubation stage, the interaction of the various active deformation mechanisms with the microstructure constituents and non-metallic inclusions play a significant role. Twinning in combination with dislocation slip within the grains has resulted in large local deformation as shown in Figure 9a,d. With further increase in deformation (see Figure 9b,e), nucleation and growth of multiple deformation twins within a single grain result in a large pileup of dislocations at GBs or twin boundaries. It can be observed from Figure 9 that, there are many instances in microstructure, where two neighboring grains which deform by twinning share a common GB or a grain which deform

by twinning share a GB with another grain which deforms by slip or three differently deformed grains share a common junction (triple points) can be observed. A large pileup of dislocations in such instances of microstructure could be observed at $\varepsilon_{>0.2}$, which could result in high stress at GBs. The stress level will be even higher if the GB lay $\perp$ to the loading direction. Such instances of microstructure could be prone to damage nucleation. The presence of inclusions such as MnS and AlN as shown in Figure 3 could also play a role at this stage. Micro-crack formation events at the MnS inclusions can be observed at $\varepsilon_{0.3}$ in Figure 9d. Many such instances of micro-cracks were also observed in AlN and MnS inclusions at very early stages of deformation at $\varepsilon_{0.10-0.3}$.

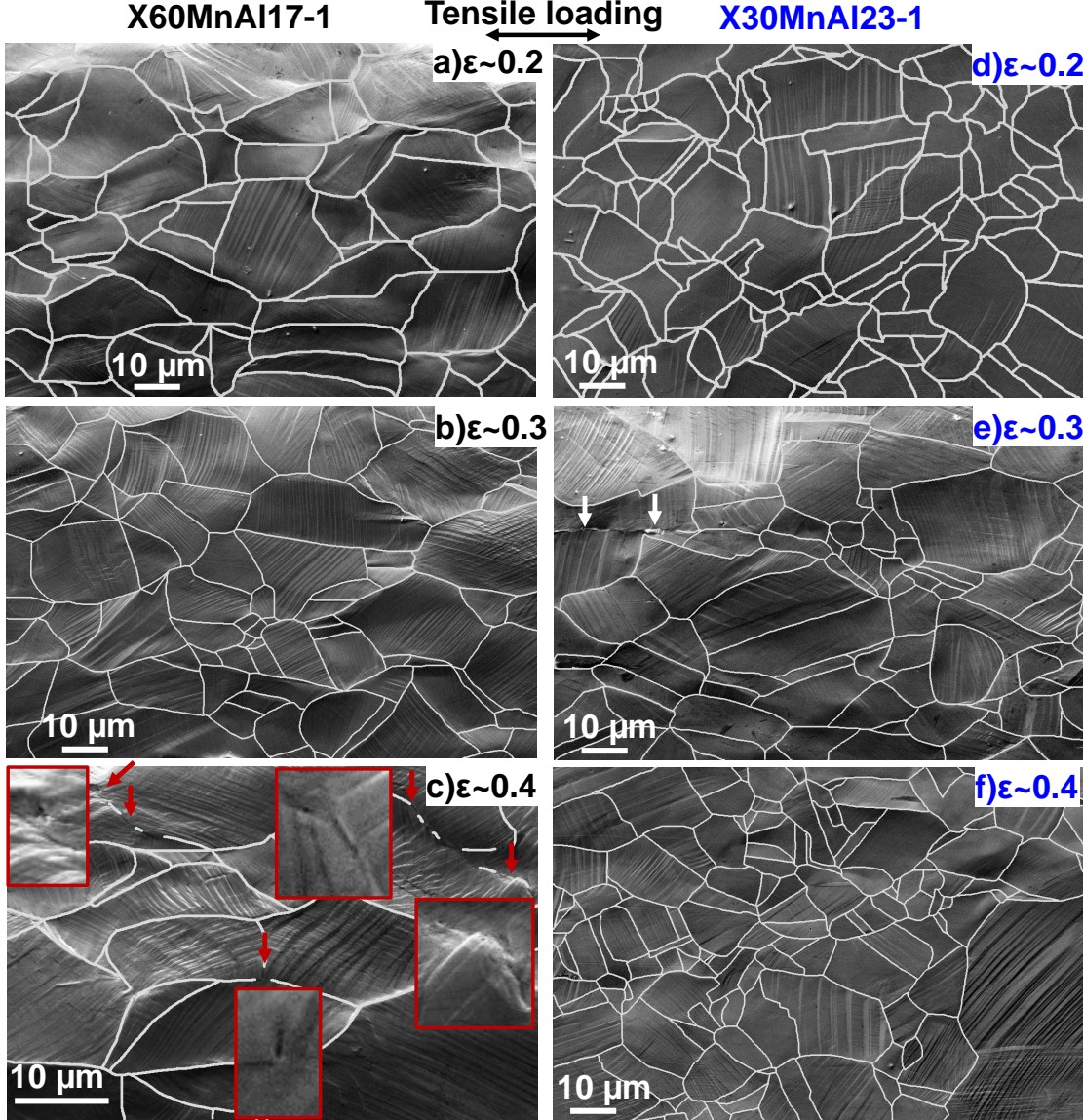

**Figure 9.** Evolution of microstructure and micro-crack formation with deformation. X60MnAl17-1: (**a**) $\varepsilon_{\sim0.2}$ (**b**) $\varepsilon_{\sim0.3}$ (**c**) $\varepsilon_{\sim0.4}$ (left column), X30MnAl23-1: (**d**) $\varepsilon_{\sim0.2}$ (**e**) $\varepsilon_{\sim0.3}$ (**f**) $\varepsilon_{\sim0.4}$ (right column); red arrows indicate micro-cracks; white arrows indicate cracks in MnS inclusions at grain boundary (GB); macroscopic true strain is indicated at the top right corner.

The saturation of the twinning within the microstructure is the precursor to the initiation of damage nucleation stage. Both the alloys deform by dislocation slip and twinning as shown in Figures 5 and 9. However, more pronounced twinning can be observed at earlier strains in X60MnAl17-1 alloy compared to an X30MnAl23-1 alloy. From Figure 5, it is very clear that at $\varepsilon_{0.3}$ large deformation due to slip and twinning within the grains can be observed in X60MnAl17-1 alloy, which

increases stress at the GB. In such high stressed GBs shown in Figure 9c micro-voids nucleate and coalesce together leading to micro-crack formation. The micro-cracks are mainly observed at GBs, triple junctions and especially in boundaries which are $\perp$ to loading direction as shown in Figure 10a,b. Thus it is very clear that stress concentration caused by the intersection of slip bands at GB lead to intergranular micro-crack formation. Many instances of micro-cracks at triple points can also be observed. However, in X30MnAl23-1 alloy the nucleation of micro-cracks did not start at $\varepsilon_{\sim 0.4}$, but at $\varepsilon_{necking}$. Many instances of inter-granular micro-crack formation events can be observed as shown in Figure 10c,d. The majority of cracks observed in the microstructure are at GB and triple points. The enlarged regions of microstructure in Figure 10 clearly indicates that micro-cracks initiated at the interception of slip bands and deformation twins at GB. From Figure 10a–d, it can be observed that the mechanism of micro-crack nucleation in both the alloys is the same. The micro-cracks which nucleated at the MnS and AlN inclusions did not grow much even after deforming to $\varepsilon_{>0.4}$ and does not propagate to the matrix (see Figure 11). Such micro-cracks can be observed in the AlN inclusions present at GB in an X60MnAl17-1 alloy as shown in Figure 11a. Similarly, in an X30MnAl23-1 alloy, micro-cracks were also observed at the MnS inclusion present at GB shown in Figure 11b and within the grain in Figure 11c. These micro-cracks were within the inclusion even after large deformation. Thus it can be stated that, micro-cracks nucleated in all inclusions present in the material, much before the beginning of the necking in both the alloys. The coalescence of micro-cracks at the inclusions present at different locations in the material could play a vital role in macroscopic failure initiation. The micro-cracks in X60MnAl17-1 alloy initiated earlier compared to X30MnAl23-1 alloy. The activation of DSA in an X60MnAl17-1 alloy could have led to an early start of damage nucleation.

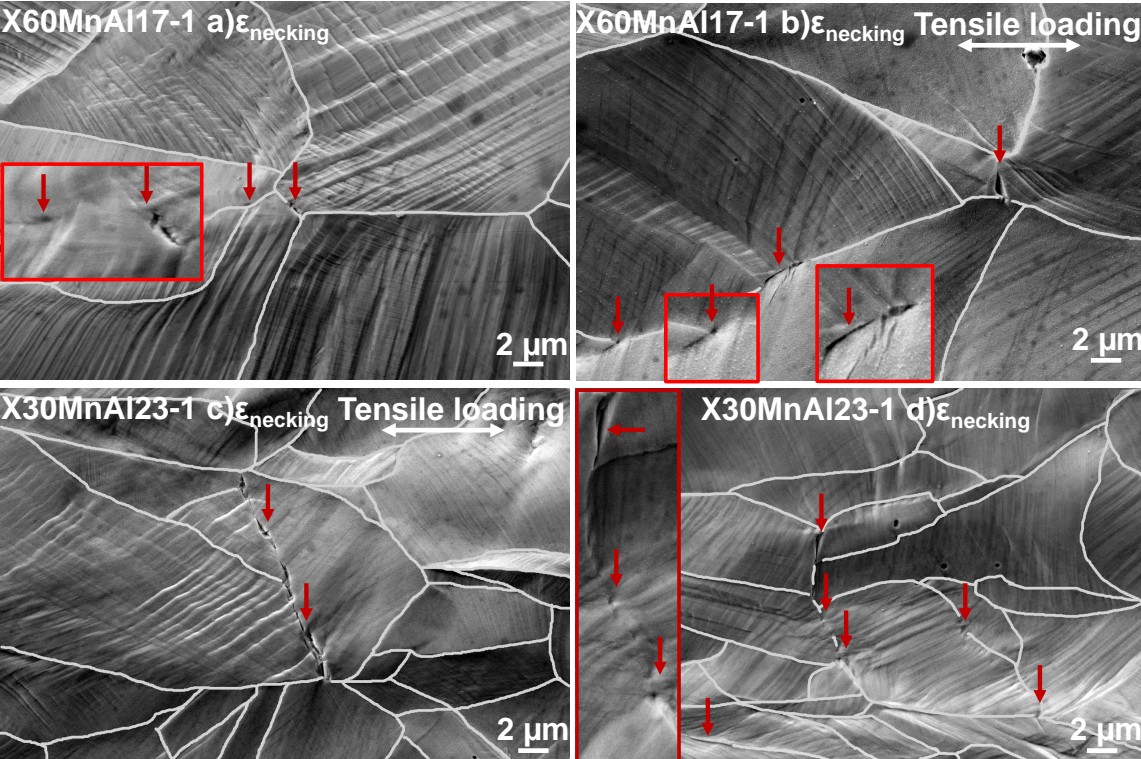

**Figure 10.** Mechanism of micro-cracks development at a strain of $\varepsilon_{necking}$. X60MnAl17-1 (**a**) Micro-crack initiation at triple junction and crack propagation into the neighboring grains assisted by the intercepting deformation twins, (**b**) Intergranular cracks nucleating at GB, X30MnAl23-1: (**c**) Micro-cracks initiation at GB and their propagation, (**d**) Intergranular cracks nucleation at grain boundaries and triple junctions; red arrows indicate micro-cracks.

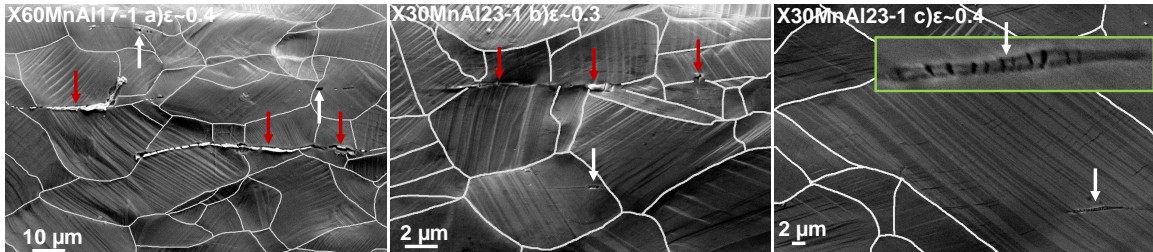

**Figure 11.** Micro-cracks at inclusions. X60MnAl17-1: (**a**) Micro-cracks in AlN inclusions at GB and in MnS inclusions close to GB and also inside the grains, X30MnAl23-1: (**b**) Micro-cracks in MnS inclusion at GB, (**c**) Micro-cracks in MnS inclusion inside a grain; red arrows indicates cracks in inclusions at GB; white arrows indicate cracks in inclusions inside the grains.

The damage growth stage starts with the coalescence of micro-cracks at various constituents of the microstructure in the material leading to macroscopic failure initiation. From the observations shown in Figures 10 and 11, it is very clear that there is a great interplay between the deformation twins and slip bands at micro-scale until the saturation of twinning. The saturation of twinning within the microstructure has led to the nucleation of micro-cracks at the high stressed GB and triple junctions. Since deformation by twinning and slip is quite uniform in almost all grains in the entire microstructure, the cracks were initiated all over the cross-section. The micro-cracks formed at MnS and AlN inclusions in the material also play a significant role in the damage growth. The variety of micro-cracks formed at different microstructure constituents coalescence together rapidly to develop macro-cracks leading to localized necking. It can be understood from the $\sigma$–$\varepsilon$ curve in Figures 4 and 6 that in both alloys large strain localization in the material occurred resulting in necking and failure. The appearance of localized necking before the failure shows that there is a plastic localization before the final failure as shown in Figure 12. This indicates that nucleation of micro-voids in the material is followed by the coalescence and growth of many micro-voids leading to micro-crack formation. The role of MnS and AlN inclusions on the failure initiation can be visualized on the fracture surfaces. The presence of large elliptical or round voids (white arrows) shows that coalescence of many smaller voids has occurred at the MnS or AlN inclusions. The large number of minute voids of size ≤2.5 µm can be seen on the fracture surfaces for both alloys. Thus it is clear that the predominant fracture mode in both alloys is a ductile failure with the formation of fine dimples.

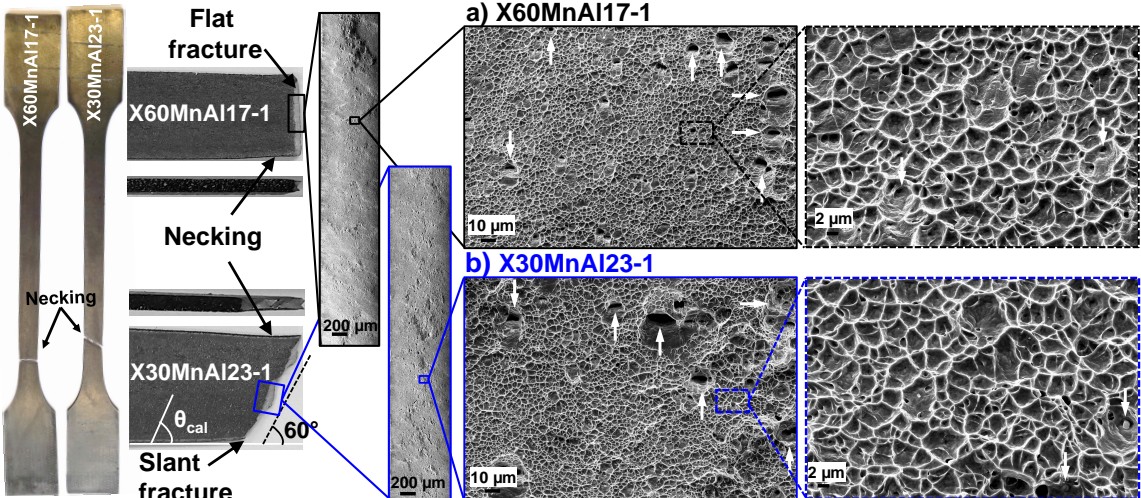

**Figure 12.** Fracture behavior of tensile specimens. Diffuse necking and fracture surface appearance (**left**) and SEM images of the fracture surfaces at different magnifications (**right**). Average dimple size of both X60MnAl17-1 and X30MnAl23-1 is approximately ≤2.5 µm. White arrows indicate large elliptical voids at MnS and AlN inclusions.

## 4. Discussion

### 4.1. Serrated Flow and PLC Effect

In an X60MnAl17-1 alloy repeated serrations can be observed on the $\sigma-\varepsilon$ curves during the plastic deformation, whereas in an X30MnAl23-1 alloy, no serrations can be observed at RT and quasi-static $\dot{\varepsilon}$ (see Figure 4). The manifestations in $\sigma-\varepsilon$ behavior in the form of serrations such as a change in mechanical properties and negative SRS is mainly due to DSA. The mechanism of DSA in C containing high Mn TWIP steels can be attributed to the formation of Mn-C short-range order or short-range cluster (SRO/SRC) [11,22,24]. It is proposed that DSA occurs when the C atoms in the Mn-C octahedral clusters reorient themselves in the core of the dislocations, thereby locking the dislocations leading to high dislocation density [36]. During this dynamic interaction between the diffusing atoms and the mobile dislocations during plastic flow, C atoms pin strongly to the dislocations increasing their pile up. When the applied stress is sufficiently high, the dislocations overcome the obstacles in the form of SRO/SRC at once by a single diffusive jump of C atoms and move to next obstacle, where they are stopped again and the process is repeated [22]. The formation of Mn-C octahedral clusters result in increased lattice resistance for dislocation glide as the passage of partial dislocation will change the local position of both substitutional and interstitial atoms [20].

The serrations were initiated at a $\varepsilon_c$ of ~12.5% in an X60Mn22 alloy without Al [14], whereas in alloys with 1.5 wt.% Al-addition, the serrations appeared at a $\varepsilon_c$ of ~24% in X60MnAl17-1 and serrations are completely suppressed in X30MnAl23-1 at RT and quasi-static $\dot{\varepsilon}$ (see Figures 6 and 7). In the Fe–Mn–Al–C alloys, the serrations are reported to occur mainly in the steels with C content above 0.6 wt.% at RT tensile testing [14,24,27–29,37]. It was stated that with the addition of Al, the diffusivity of the carbon will be reduced and the serrated flow will be shifted to higher temperatures [38]. The addition of Al has suppressed the DSA phenomena by increasing the activation energy for C diffusion and reducing the interaction time between stacking faults and the Mn-C SRO [16]. However, even in Al-added alloys, the DSA could occur at elevated temperatures because of the increased defect mobility resulting in serrated $\sigma-\varepsilon$ curves.

Twinning behavior in both Al-added alloys are quite similar, except for the initiation and saturation of twinning and TVF (see Figures 5 and 13). Even though twinning occurs in both alloys, the serrated $\sigma-\varepsilon$ flow is observed only in X60MnAl17-1 alloy and not in X30MnAl23-1 alloy. The local deformation behavior is inhomogeneous in an X60MnAl17-1 alloy due to the initiation and propagation of deformation bands during deformation commonly known as Portevin-LeChatelier (PLC) effect, whereas in an X30MnAl23-1 alloy homogenous $\sigma-\varepsilon$ behavior can be observed. The nucleation and propagation of deformation bands result in serrations on the $\sigma-\varepsilon$ curves which can be ascribed to the DSA. Kang et al. showed that the number of Mn-C SRO clusters increases with increasing macroscopic strain in X60Mn18 alloy [23]. Song and Houston reported that with an increase in $\varepsilon$, the volume fraction of Mn-C SRO/SRC increases and the mean radius of clusters decreases [24]. Madivala et al. investigated an X60Mn22 alloy and showed that serrations disappear at elevated temperatures and at high strain rates. It was also stated that twinning occurs at these temperatures and at high strain rate tests [14,19]. Koyama et al. also indicated that the deformation twinning and martensitic transformation did not cause the serrations [39]. Lebedkina et al. suggested that twinning starting at some point in the specimen will initiate twins in the neighboring grains and advance into the unoccupied regions of the specimen. However, they claimed that twinning cannot alone responsible for the entire strain accumulation during strain jumps due to low TVF observed in TWIP Steel [40]. Recently, Sevsek et al. proposed a multi-scale-bridging model for the formation and propagation of the deformation bands and the resulting occurrence of the serrated flow in high Mn TWIP steels. However, they also stated that neither the SRO-based theory nor the deformation twinning based theory proposed could explain the experimental observations completely [37]. Based on the experimental observations from this study and the from the literature, it can be clearly stated that deformation twinning and $\varepsilon$-martensite

transformation cannot be responsible for causing serrations in Fe-Mn-C-Al alloys. Thus, DSA is the dominant factor for causing the serrations in an X60MnAl17-1 alloy.

The strain localization within the deformation bands in X60MnAl17-1 alloy is slightly different over the gauge length during the transition regime (see Figure 7l–n). After their complete development, the strain localization in them increases drastically compared to other areas (see Figure 7n). The $\varepsilon_{local}$ is much larger in the region where deformation band has already passed by (see in Figure 7m,n marked by 'B', 'D') compared to another region where the deformation bands has to pass through (see in Figure 7m,n marked by 'A', 'C'). The velocity of propagation of the deformation bands decreases with increase in macroscopic strain. Song and Houston showed by SANS experiments that the size and number density of SRO is much larger in the regions where deformation band has already passed by compared to the region where it has to pass through [24]. Also due to the fact that SRO/SRC offers higher resistance to dislocation glide thereby leading to (a) aging of dislocations by C-atoms during their interaction time at obstacles (b) enhancement of diffusion coefficient by vacancies generated during plastic flow and (c) increase in mobile dislocation density with strain. This results in increased resistance offered by the Mn-C clusters and also by the deformation twins for the bands propagation leading to decrease in velocity and difference local deformation behavior.

## 4.2. Strain Hardening and Twinning Evolution

Deformation twinning during plastic deformation plays a significant role in enhancing the strain hardening behavior of materials through the creation of new boundaries thereby subdividing the original grains commonly known as dynamic Hall-Petch effect [10,13,39]. Twin boundaries act as the barriers to dislocation motion thereby progressively reducing the effective mean free path (MFP) of dislocations resulting in enhanced strain hardening. Bouaziz et al., De Cooman et al., Koyama et al. claim that mechanical twinning and the related dynamic Hall-Petch effect associated with TWIP effect remain the only mechanisms that can explain the high strain hardening of TWIP steel [10,13,39]. However, the role of DSA in manifesting the SHR of TWIP steels as explained in Section 4.1 cannot be neglected.

An X60MnAl17-1 alloy, where DSA can be observed shows enhanced strain hardening and ductility compared to an X30MnAl23-1 alloy, where DSA does not occur (see Figure 13). Such sustained and high work hardening rates can also be seen in austenitic steels with high C content because of DSA [41]. The ab-initio based study combined with SANS experiments by Song et al. [42] and SANS experimental study by Kang et al. proved the presence of Mn-C octahedral clusters and their evolution, indicating the occurrence of DSA during deformation in high Mn steels. The occurrence of static strain aging due to the presence of Mn-C SRO was stated to be responsible for increased yield strength and pronounced yielding of annealed TWIP steel [43]. It was stated that deformation twinning in TWIP steels is promoted by thermally activated nature of dislocation slip due to dynamic interactions between solute C atoms and dislocations that inhibit dislocation slip due to lattice friction effects [44]. It was proposed that the trapping of slowly moving partial dislocations by C atoms results in increased inter-granular stress and reduction of MFP [37,39]. The increased localized stress near the GB facilitate grain boundary nucleation of deformation twins [45]. DSA is also showed to be crystallographic orientation dependent, could occur at elevated temperatures and is enhanced by increasing C content [39]. Thus in an X60MnAl17-1 alloy, the occurrence of DSA has led to increased stresses at GBs. This led to triggering of the nucleation of multiple deformation twins in the preferably oriented grains. Whereas in X30MnAl23-1 alloy, conventional twinning mechanisms occurs in combination with dislocation glide at RT (see Figure 5). Thus the yield strength increase by static strain aging and the indirect promotion of the mechanical twinning during deformation in an X60MnAl17-1 alloy, could have led to high SHR and enhanced ductility.

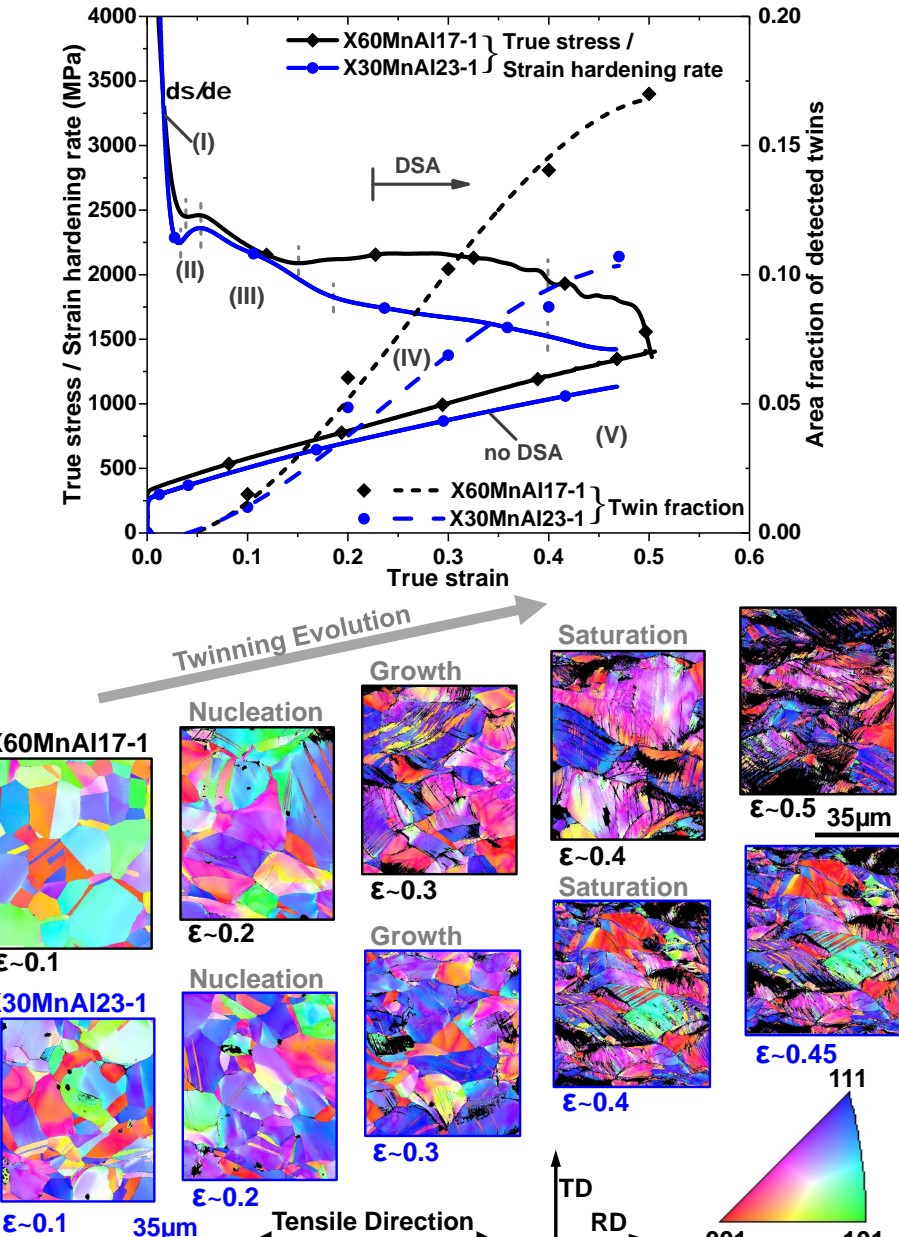

**Figure 13.** Strain hardening in relation to the evolution of twinning; strain hardening behavior (**top**) and EBSD IPF maps with increasing macroscopic true strain (**bottom**).

The evolution of twinning with increasing macroscopic strain is correlated with the SHR as shown in Figure 13. The RD-IPF maps at different strains are also shown. Based on the detailed work by De Cooman et al., Gutierrez-Urrutia and Raabe, the SHR in the current study is divided into five stages [13,46]. Stage (I) is the initial strain hardening stage controlled by the dislocation density evolution without the initiation of twins. The initial downfall of SHR is characterized by a continuously decreasing rate of dislocation storage and an increasing rate of dynamic recovery. The dislocation reactions in this stage are responsible for the nucleation of twins. Stage (II) is characterized by an increase of SHR in both alloys and is referred to as the primary twinning stage. It begins slightly above yielding and at a strain of ∼0.03 in an X30MnAl23-1 alloy and at a strain of ∼0.04 in an X60MnAl17-1 alloy (see Figure 4). The stress at the beginning of stage II is referred to as the twin initiation stress. It is about 345 MPa in an X30MnAl23-1 alloy and is about 435 MPa in X60MnAl17-1 alloy. The evolution of dislocation structures and the initiation of primary twins are supposed to be responsible for the rise of SHR in this stage. The maximum SHR for both the alloys occurs at ∼0.05 strain at the end of stage II.

Stage (III) is characterized by a decrease in SHR with increasing macroscopic strain. It can be attributed to the reduced rate of primary twins initiation. It was also stated that the initial twins subdivide the original grains thereby reducing the MFP of dislocations, which results in increased stress for twin nucleation. At the end of this stage, twins can be clearly observed in the preferably oriented grains as shown in Figure 13 at a strain of 0.2. Stage (IV) and (V) are quite different in both the alloys. Stage (IV) and (V) in an X30MnAl23-1 alloy marked by a continuous decrease in SHR, ascribed to the reduced additional refinement of the dislocation and twin substructures, together with the increasing strengthening effect of the individual twins as obstacles to dislocation glide, reduce the capacity for trapping more dislocations. Stage (IV) in an X60MnAl17-1 alloy can be distinguished by again increase in SHR and reaching maximum between a strain of 0.3 and 0.4. This stage is characterized by the activation of the secondary twin systems and the formation of the multiple twin-twin interactions. This results in a further subdivision of grains and thus reduces the MFP of the dislocations considerably leading to increasing or constant SHR. During stage (V) in X60MnAl17-1 alloy, the SHR decreases continuously. In this stage TVF increases significantly and the twin bundles get thicker and denser. This can be observed in the microstructure at a strain of 0.4 and 0.5. The major difference in SHR between two alloys can be observed at stage (III), at which DSA gets activated in X60MnAl17-1 alloy which has indirectly led to significant increase in TVF, whereas in X30MnAl23-1 alloy, the TVF increase was marginal.

### 4.3. Damage and Fracture

Heterogeneous deformation twinning and dislocation slip could be observed in the microstructure in the early stages of deformation (see Figures 5 and 13). Micro-cracks could be observed in the MnS and AlN inclusions at strain below 0.2 (see Figure 11). The nucleation and propagation of deformation bands due to DSA has resulted in localization of strain within the bands leading to failure initiation (see Figure 7). Thus twins, slip bands, deformation bands, and non-metallic inclusions play a significant role in the fracture phenomena in TWIP steels.

In both alloys, dislocation glide and deformation twinning are the major active deformation mechanisms. Even though both these deformation mechanisms did not induce damage directly, but they acted as a source for increasing the stress concentration within the microstructure, which is a precursor step for damage initiation. A schematic diagram illustrating the mechanism of micro-crack formation in TWIP steels based on the experimental observations is depicted in Figure 14. The development of large inter-granular regions with high dislocation density within the microstructure due to the subdivision of grains due to twinning and also deformation by slip resulted in pronounced inter-granular stress. Deformation by twinning and dislocation slip can be observed in the microstructure in the preferably oriented grains leading to the localization of the plastic strain. It could be observed that <111> tensile orientation is the preferential orientation for the deformation twinning and <100> is the non-preferential tensile orientation for twinning. Many bundles of primary and secondary deformation twins can be observed in <111>-oriented grains and slip bands in <100>-oriented grains (see Figure 13). Largest plastic strain concentration can occur in the <111>-oriented grains, which are preferred for the deformation twinning and the second largest strain concentration could occur in the <100>-oriented grains, which are preferred for the dislocation slip. The large plastic localization associated to accommodate the plasticity within the grains results in high-stress concentration at the GB. This results in the initiation of voids which combine together leading to initiation of inter-granular micro-cracks (see Figures 10 and 14). Even though micro-cracks initiated at inclusions at an early stage, their growth during deformation was limited (see Figure 11). Thus at a micro-scale, stress concentration caused by the interception of deformation twins and slip bands at GB lead to damage initiation. This results in the intergranular crack propagation with a fracture surface characterized by shallow dimples (see Figure 12).

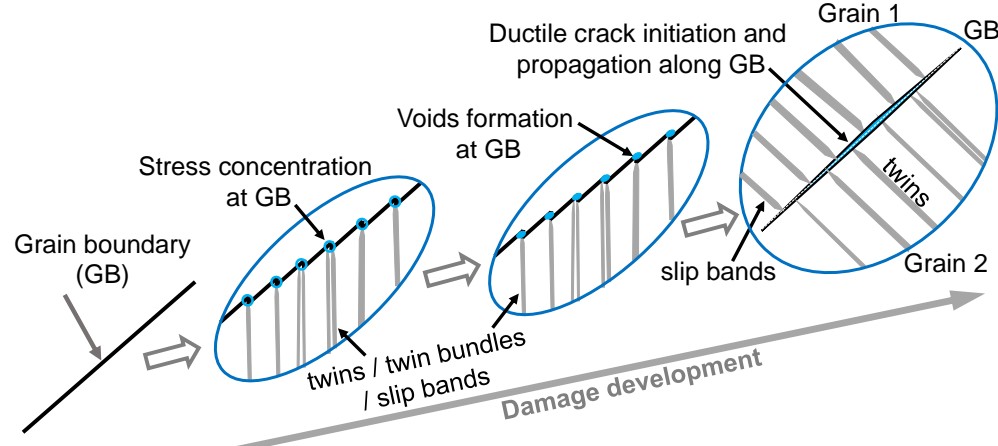

**Figure 14.** A schematic illustration of micro-cracking mechanisms in TWIP steels based on the experimental observations from interrupted micro-tensile tests carried out in the SEM. Stress concentration caused by the intercepting deformation twins at GB. Void formation and their growth leading to inter-granular micro-crack initiation at GBs.

## 5. Conclusions

The Al-added TWIP steels X60MnAl17-1 and X30MnAl23-1 were investigated by carrying out uni-axial tensile tests. The local plastic strain ($\varepsilon_{local}$) and temperature evolution during deformation were monitored in conjunction with digital image correlation and synchronous temperature measurements using thermocamera. Interrupted tensile tests were carried out in SEM along with EBSD measurements to study the microstructure evolution and micro-cracks development. The main conclusions can be drawn as follows:

- Strain hardening rate of an X60MnAl17-1 alloy is extraordinarily high compared to an X30MnAl23-1 alloy. An X60MnAl17-1 alloy showed higher yield strength, tensile strength and elongation compared to an X30MnAl23-1 alloy. The enhanced mechanical properties of the X60MnAl17-1 alloy is mainly due to the enhanced deformation twinning in addition to dislocation glide and also activation of dynamic strain aging (DSA). DSA is completely suppressed in an X30MnAl23-1 alloy at room temperature and quasi-static strain rate due to lower carbon content.

- Twining is the most predominant deformation mechanism occurred along with dislocation slip in both the alloys. The addition of Al has led to increased stacking fault energy thereby delaying nucleation of deformation twins and prolonged the saturation of twinning.

- Micro-cracks are observed at elongated MnS inclusions or at AlN inclusions at a relatively small strain of ∼2/3 of total strain. However, these micro-cracks showed no tendency to grow.

- Large heterogeneous deformation within the grains by twinning or dislocation slip has led to a high-stress concentration at grain boundaries (GBs) due to the interception of deformation twins and slip band extrusions at GBs. Hence micro-cracks in Al-added TWIP steels originated mainly at grain boundaries and triple junctions.

- In an X60MnAl17-1 alloy, the occurrence of DSA has led to inhomogeneous flow behavior due to the nucleation and propagation of deformation bands during deformation. This resulted in large strain localization within the deformation bands and the velocity of band motion decreased with increasing strain due to the intersection of two bands. The $\varepsilon_{local}$ accumulation within the intersecting bands resulted in a macroscopic crack initiation close to the edges of the tensile specimen. In an X30MnAl23-1 alloy, homogenous $\varepsilon_{local}$ distribution throughout the gauge length could be observed until the beginning of necking. Thereafter failure in the material occurred by classical necking and strain localization.

- The ductile failure mode is the most predominant mode of failure in Al-added TWIP steels, mainly characterized by the formation of very fine dimples with a crack propagation along GBs.

**Author Contributions:** M.M. designed, performed and analyzed the experimental data such as SEM, tensile tests and mechanical properties. A.S. performed the EBSD measurements and analyzed the data. W.B. and U.P. contributed with ideas and intensive discussions. M.M. wrote the initial draft version. All authors contributed equally to the interpretation of results and writing the final version of the manuscript.

**Funding:** This research was funded by the German Research Foundation (DFG) within the framework of the SFB 761 "Steel-ab initio".

**Acknowledgments:** This work was carried out within the framework of the SFB 761 "Steel-ab initio" under the sub-project C6: Damage and failure. We would like to thank all the members of the SFB 761 "Steel-ab initio" project for their valuable cooperation and tremendous support.

**Conflicts of Interest:** The authors declare no conflict of interest.

## Appendix A. Macro-Tensile Tests Setup for the Local Strain Analysis Using Digital Image Correlation

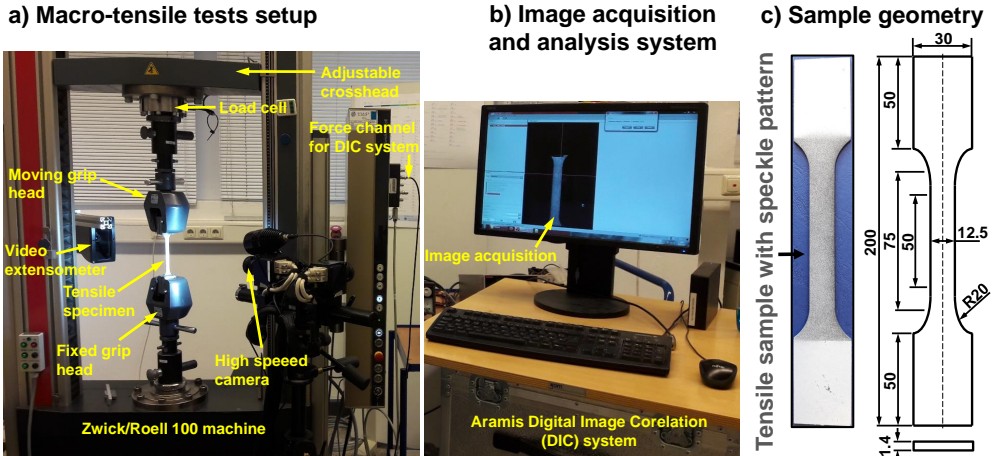

**Figure A1.** Macro-Tensile tests carried out in conjunction with DIC for investigating the local plastic strain ($\varepsilon_{local}$) evolution, nucleation, and propagation of deformation bands with increasing strain. (**a**) Tensile testing machine depicting the adjustable crosshead, grips for clamping, load cell, video extensometer and high-speed camera for image acquisition, (**b**) Aramis DIC system used for analysis, (**c**) Tensile sample with black and white speckle pattern prepared for $\varepsilon_{local}$ measurement.

## Appendix B. Macro-Tensile Tests Setup for Temperature Measurement using High Resolution Thermocamera

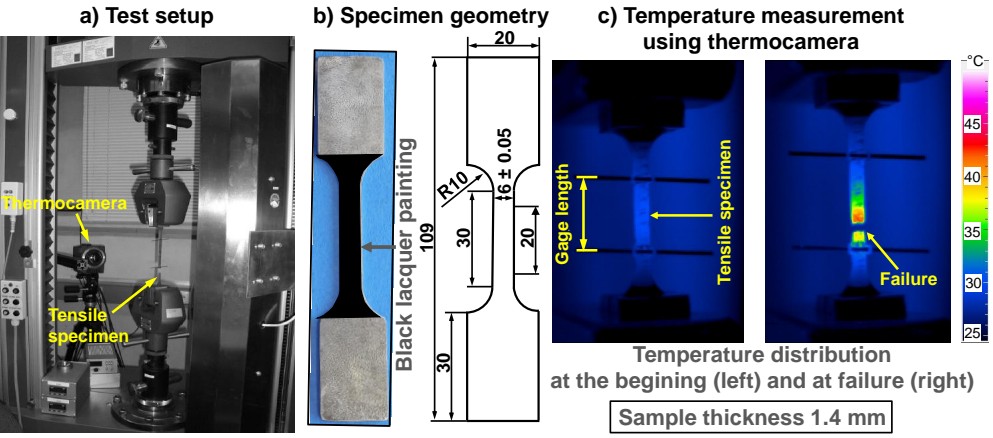

**Figure A2.** Temperature measurement setup using Macro-Tensile tests carried out in conjunction with high-resolution thermocamera for investigating the rise in temperature due to adiabatic heating. (**a**) Tensile testing machine depicting the thermocamera, tensile sample, (**b**) sample painted with black lacquer painting to minimize the reflections from the surroundings, (**c**) Image acquisition and analysis using IRBIS® online software.

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
