# Peer review of "Strain Hardening, Damage and Fracture Behavior of Al-Added High Mn TWIP Steels"

_metals, doi:10.3390/met9030367_

Reviewer 1 Report

The experimental study was run carefully, using various, up to date techniques, but the the results (and their discussion) are not very surprising nor original (for example, what else than ductile fracture with broken inclusions and dimples did the authors expect after more than 50% strain? Only  intergranular damage is relatively new). Besides,  the paper is way too long and verbose, with many repeated (and sometimes not rigorous enough) comments. For a better impact, its length  should (and might easily) be reduced  by 1/3 to 1/2 (reduce in particular the abstract and long discussion of page 21. Reduce the comments on ductile fracture and remove  fig 12. The decomposition in successive hardening stages has already been made in many papers, and might be skipped as well ) .

On the other side, some experimental details are missing:

In page 4, the voltage and duration for electropolishing are specified, but not the solution....

The gage length and width of the tensile specimens (twice in page 5) are specified, but not their thickness...It is not clear whether the strain rate is actually controlled, or if it is the displacement rate. Is a  load recording frequency of 1Hz sufficient to capture well the  serrations?  The serrated regime is a nuisance for a correct derivation of the strain hardening rate. How was this done (preliminary smoothing of the stress-strain curves or other technique?)?

The materials grain size and shape are  presented, but no element  are provided about  their textures.

Explain what is meant by ECO-index. Explain how the Lankford coefficient was obtained.

On page 8 (line 244) it is stated that "mechanical twins are not initiated in both alloys at a strain of 0.1". This is contradicted on page 18 (line 527) where the onset of twinning is said to be at 0.00313 and 0.0375 (not a reasonable  precision!) . The problem seems to be that the areas monitored by SEM / EBSD are too small to constitute a representative volume element, which is a serious limitation, and the 100 nm step used for EBSD does not allow an accurate detection of the onset of twinning.

Fig 9 is not a DIC strain map and should not be commented in terms of strain homogeneity/heterogeneity (bottom of page 12, for example).

The 5 mJ/m2 difference in SFE of the two alloys (24 and 29 J/m2) is considered neglibible (it is even described as similar SFE materials in the abstract, which should be corrected), but a adiabatic heating-induced rise in SFE by 2.5mJ/m2 (the way it is computed is not specified)  is considered significant... Furthermore, this rise in SFE is considered to favor twinning, which is contrary to the common knowledge about the evolution of the critical shear stress for twinning with the SFE (e;g. Venables). The stress-induced variations of SFE are probably much more significant than those from heating.

A single strain rate was used for the tensile tests. Clearly, this is not enough to claim as the authors repeatedly do that DSA is "completely suppressed at RT" in one of their alloys. All these sentences should be rephrased in a more careful style mentioning the strain rate at which no serrations were observed.

Grain boundary sliding is evoked, but unless more convincing elements are shown to support it, it should be removed from the text and from fig 15.

At several places, DSA is presented as a deformation mode by itself, "in addition to dislocation glide and twinning". This is not correct, since DSA is not a deformation mechanism (it rather restrain plastic flow) but a characteristic of dislocation glide.

The idea of DSA-assisted twinning "in addition to conventional twinning" should not be presented as a straightforward conclusion issued from the presented data, but more carefully, as an interpretation. The fact that in most materials DSA actually reduces ductility, while in the paper it is supposed to indirectly increase it,  by triggering twinning  should be recalled and  discussed.

All the observed differences between the two alloys are attributed to their different Al contents, while they also differ substantially in Mn, Si and C contents. Again, the comments should be more careful.

The authors mention "saturation of deformation" (page 14, line 363, but some other places as well), while they should just mention saturation of twinning, because obviously the material continues to deform plastically until failure...

Ref 32 and 34 are twice the same

+- error range should be added for the data in Table 3 .

Author Response

The authors express their deepest thanks to the reviewer and editors for their valuable comments and suggestions. We have striven to incorporate the suggested changes. A summary of the changes made is provided below.

Reviewer 2 Report

The title suggests that the paper deals with Al effect. However, the effect of Al is not directly studied.

GB appears in the abstract before explanation.

Mistypings like "It was showed..." (line 63), "Elastic properties ... was ..." (line 175+), etc., could be corrected.

The end of abstract looks confusing. "...Al-addition has led to .... suppression of serrated flow caused due to dynamic strain aging (DSA) ... However, an alloy with DSA showed enhanced strain hardening and ductility compared to an alloy without DSA ... Thus, Al-added TWIP steels not only showed superior mechanical properties but also exhibited excellent resistance to damage."

The meaning of "saturation of deformation" is not clear.

The statement of "...the twinning in X60MnAl17-1 alloy saturates much earlier compared to an X30MnAl23-1 alloy" (lines 365-366) contradicts Fig. 14, which does not suggest any difference in saturation behavior.

The statement that DSA promoted the deformation twinning is not proved. These two phenomena take place concurrently, but might have different origins.

The papers by De Cooman et al., Gutierrez-Urrutia and Raabe (line 521), could be mentioned with reference numbers.

Regarding conclusion 1. SHR was higher in the 0.6C alloy than that in the 0.3C alloy before DSA started to operate.

Regarding conclusion 2. It is generally agreed that deformation twinning occurs in low SFE FCC-alloys. Therefore, an increase in SFE should not promote the twinning.

Author Response

(The authors gave the same response as above.)

Reviewer 3 Report

The authors report a systematic study of strain hardening and damage behavior of Al-added steels. The study is focused on comparing two different alloying concepts by varying C and Mn contents. Experiments have been accurately carried out. The conclusions are well supported by the reported results. The article is well written. I have enjoyed reading it. My opinion is that the article should be published. I have only suggestions of minor changes which could be of help for improving the manuscript.

1.       In the introduction please add a reference to support: “Through many years of development and application, advanced high strength steels (AHSS) has proved themselves to be versatile and effective materials for automotive parts.”. For instance; Pekka Kantanen, Mahesh Somani, Antti Kaijalainen, Oskari Haiko, David Porter and Jukka Kömi, Metals 2019, 9(2), 256.

2.       I suggest to also modify the sentence by mentioning “and metallic alloys” after (AHSS) citing for instance works on titanium alloys: D Smith, O P J Joris, A Sankaran, et al., Journal of Physics: Condensed Matter 2017, 29, 155401. By doing these, the article will be appealing to a broader audience and not only to those interested in steels.

3.       Mention the accuracy in determining chemical compositions, which is the minimum content you can detect?

4.       Make sure you have defined all the acronyms used.

5.       True strain is dimensionless, replace (-) by (dimensionless).

6.       For number fraction (%) should be used instead of (-).

7.       Density, young modulus, shear modulus, and Poisson’s ratio should be given with errors.

Author Response

(The authors gave the same response as above.)

Reviewer 4 Report

This work deals with the study of the behaviour of TWIP steels, from the viewpoint of strain hardening, damage and fracture analysis. Specifically the evolution of microstructure, deformation mechanisms and micro-cracks was investigated by increasing the imposed deformation.

The work is carried out by comparing the analysis provided by different experimental experimental techniques. Even the research is very interesting, in my opinion some parts require more enhancements in order to increase the soundness of the research.

1. INTRODUCTION

The introduction is very complete under the point of view of material peculiarities description and the phenomena involved in applying deformation to TWIP steels, but I find it lacks of a state of the art of experimental techniques studied and presented in this work.

In literature there are very well-known experimental techniques adopted for measuring damage and crack behaviour of ferrous and non-ferrous metals, such as: Optical Microscope, Magnetic flux density, Digital Imaging Correlation, Thermoelastic Stress Analysis… Following this some interesting paper referring to novel applications are reported below:

•        F. Mathieu, F. Hild, S. Roux. Image-based identication procedure of a crack propagation law. Engineering Fracture Mechanics, 2013. DOI: 10.1016/j.engfracmech.2012.05.007

•        Fedorova, A.Yu., Bannikov, M.V., Plekhova, E.,V., Plekhov O.A., 2012, Infrared thermography study of the fatigue crack propagation Fracture and structural integrity 21, 46-53.

•        D. Palumbo, R. De Finis, F. Ancona, U. Galietti. Damage monitoring in fracture mechanics by evaluation of the heat dissipated in the cyclic plastic zone ahead of the crack tip with thermal measurements. Engineering Fracture Mechanics, 181 65-76, 2017

•        Ancona F, De Finis R, Palumbo D, Galietti U,  Crack Growth Monitoring in Stainless Steels by Means of TSA Technique. Procedia Engineering, 109, 89-96, 2015.

•        Réthore J, Limodin N, Buffière JY, Roux S, Hild F. Three-dimensional analysis of fatigue crack propagation using X-Ray tomography, digital volume correlation and extended finite element simulations. Proc IUTAM 2012;4(4):151–8.

•        Tanabe H, Kida K, Takamatsu T, Itoh N, Santos EC. Observation of Magnetic Flux Density Distribution around Fatigue Crack and Application to Non-Destructive Evaluation of Stress Intensity Factor. Proc Eng 2011;10:881–7.

•        Yates JR, Zanganeh M, Tai YH. Quantifying crack tip displacement fields with DIC. Eng Fract Mech 2010;77:2063–76.

•        Guduru PR, Zehnder AT, Rosakis AJ, Ravichandran G. Dynamic full field measurements of crack tip temperatures. Eng Fract Mech 2001; 68:1535–56

 The addition of these papers is required also because the authors use digital imaging correlation and thermography to study the damage behaviour of TWIP steels.

Please declare clearly the aims of the work and the novelty of the same as well as its impact on scientific scenario.

2. MATERIALS AND METHODS

The author present in this paper two types of tensile tests: micro-tensile and ‘macro-tensile’ that is the well-known tensile tests on standard samples. It is very interesting, so that I suggest to well define these two type of tests by presenting the samples and by declaring the different outcomes from such the analysis.

The author stated that the stacking fault energy was calculated according ref.4. How did the author calculate the stacking fault energy? Please provide more explanations.

Please provide graphically the geometry and dimensions of samples for tensile testing.

Please resume in a table the strain values adopted, explaining why have been chosen that values and why such the increment between a strain and another?

Experimental setups have to be shown in the manuscript.

Why the authors choose a quasi-static strain rate, please clarify?

3. RESULTS

The author for detecting the local deformation behaviour used Digital Imaging Correlation. Please provide a description of setup in the appropriate section, which was the adopted high-speed camera? Experimental setup? How did the author realize the speckle pattern on sample?

In figure 8b please add the color-bar.

Which was the geometric resolution of thermographic measurements? That is, which was the adopted mm/pixel ratio?

Where did the fracture of samples occur?it is important for a better reading of thermograms.

By monitoring the temperature during tensile test, it is possible to draw some other considerations on mechanical behaviour of sample: its fatigue limit, the thermoealstic inversion point that is the limit for reversible deformations. Please refer to: Luong. Infrared thermographic scanning of fatigue in metals. Nuclear Engineering and Design 158 (1995) 363-376.

Author Response

The authors express their deepest thanks to the reviewer and editors for their valuable comments and suggestions. We have striven to incorporate the suggested changes. A summary of the changes made is provided below.

Round  2

Reviewer 1 Report

The authors took into account a substantial fraction of the remarks, but not all:

* the paper still mentions GB sliding in the text (p21) and in Fig 15, while sliding  is not visible on Fig 10, where only GB micro-cracks can be seen. Shifted scratch lines or fiducial marks accross the GBs would actually be necessary to prove sliding). It is still recommended to rephrase lines 579-581 on page 21

* An increase in SFE is still assumed to trigger twinning (line 524 on p18) , which is contrary to accepted trends (Venables, Byun...).

* although they claim that the paper was substantially shortened, the paper is now 25 pages long instead of 24 !  No effort to make the paper more concise was done. The advice to shorten the paper was given in its own interest, since readers do prefer short and clear papers!

* in the last line of the  abstract, "mechanical properties" , which does not evoke only  tension, but also other aspects like fatigue or stress-corrosion cracking resistance should be replaced by "tensile properties".

Author Response

The authors express their deepest thanks to the reviewer for his/her valuable comments and suggestions. We have striven to incorporate all the suggested changes.

Author Response

(The authors gave the same response as above.)
